# Tet inactivation disrupts YY1 binding and long-range chromatin interactions during embryonic heart development

Shaohai Fang[1,11], Jia Li[1,11], Yang Xiao[2,3], Minjung Lee[1], Lei Guo[1], Wei Han[1], Tingting Li[1], Matthew C. Hill [4], Tingting Hong[1], William Mo[1], Rang Xu[5], Ping Zhang[5], Fen Wang[6], Jiang Chang[6], Yubin Zhou [6,7], Deqiang Sun [1], James F. Martin[2,3,4,8] & Yun Huang [1,9,10]

Tet-mediated DNA demethylation plays an important role in shaping the epigenetic landscape and chromatin accessibility to control gene expression. While several studies demonstrated pivotal roles of Tet in regulating embryonic development, little is known about their functions in heart development. Here we analyze DNA methylation and hydroxymethylation dynamics during early cardiac development in both human and mice. We find that cardiac-specific deletion of Tet2 and Tet3 in mice (Tet2/3-DKO) leads to ventricular non-compaction cardiomyopathy (NCC) with embryonic lethality. Single-cell RNA-seq analyses reveal a reduction in cardiomyocyte numbers and transcriptional reprogramming in cardiac tissues upon Tet2/3 depletion. Impaired DNA demethylation and reduced chromatin accessibility in Tet2/3-DKO mice further compromised Ying-yang1 (YY1) binding to its genomic targets, and perturbed high-order chromatin organization at key genes involved in heart development. Our studies provide evidence of the physiological role of Tet in regulating DNA methylation dynamics and chromatin organization during early heart development.

---

[1] Center for Epigenetics & Disease Prevention, Institute of Biosciences and Technology, Texas A&M University, Houston, TX 77030, USA. [2] Texas Heart Institute, Cardiomyocyte Renewal Lab, Houston, TX 77030, USA. [3] Department of Molecular Physiology and Biophysics, Baylor College of Medicine, One Baylor Plaza, Houston, TX 77030, USA. [4] Program in Developmental Biology, Baylor College of Medicine, One Baylor Plaza, Houston, TX 77030, USA. [5] Xinhua Hospital, School of Medicine, Shanghai Jiao Tong University, Shanghai, China. [6] Center for Translational Cancer Research, Institute of Biosciences and Technology, Texas A&M University, Houston, TX 77030, USA. [7] Department of Medical Physiology, College of Medicine, Texas A&M University, College Station, TX 77843, USA. [8] Cardiovascular Research Institute, Baylor College of Medicine, Houston, TX 77030, USA. [9] CPRIT Scholar in Cancer Research, Houston, USA. [10] Department of Molecular & Cellular Medicine, College of Medicine, Texas A&M University, College Station, TX 77843, USA. [11]These authors contributed equally: Shaohai Fang, Jia Li. Correspondence and requests for materials should be addressed to D.S. (email: dsun@tamu.edu) or to J.F.M. (email: jfmartin@bcm.edu) or to Y.H. (email: yun.huang@tamu.edu)

The Ten-Eleven Translocation (TET) protein-mediated DNA modification pathway plays an important role in regulating DNA methylation and demethylation homeostasis during development[1–3]. Deletion of all three TET genes (TET1-3) impairs differentiation of both human and mouse embryonic stem cells (ESCs)[4,5]. Furthermore, Tet-triple deficient mice exhibits developmental defects at the gastrulation stage[2], indicating an indispensable role of Tet enzymes in early embryonic development. TET belongs to the $Fe^{2+}$ and 2-oxoglutarate-dependent dioxygenase family that successively oxidizes 5-methylcytosine (5mC) to 5-hydroxymethylcytosine (5hmC), 5-formylcytosine (5fC) and 5-carboxylcytosine (5caC)[6–9]. 5hmC is one of the most abundant and relatively stable modifications among all these oxidized forms of DNA methylation[7,8]. TET-catalyzed DNA hydroxymethylation is enriched at enhancer and open chromatin regions during cellular differentiation and embryonic development[10–13], thereby pointing to yet-to-be-clarified functions of Tet/5hmC in chromatin biology and gene regulation. Somatic mutations of TET2 are frequently detected in individuals with clonal hematopoiesis, which are closely associated with high risk of cardiovascular disease[14–16].

Cardiac differentiation during embryonic development is tightly regulated through precise control over gene expression, when cells receive a multitude of intracellular and extracellular cues[17]. Epigenetic factors, such as DNA modifying enzymes DNMTs, play indispensable roles in choreographing this exquisitely coordinated process by directly participating in the programming of cardiac transcriptional networks, thereby exerting control over gene expression to orchestrate early heart development[18–20]. Aberrant epigenetic modifications arising from genetic alterations in these key enzymes and/or environmental risk factors, such as folate deficiency, may cause developmental defects in the heart and potentially lead to embryonic lethality in mice, as well as human cardiomyopathies[21–23]. A deeper understanding of epigenetic regulatory mechanisms that modulate cardiac gene expression is crucial for deciphering the molecular etiology of congenital heart defects.

In this study, we systematically investigated the DNA methylation and hydroxymethylation dynamics during early cardiac development in both human and mice. We generated a cardiac-specific Tet2 and Tet3 double deficient mouse model to investigate the function of Tet-mediated DNA modifications during early cardiac development. These mice developed non-compaction cardiomyopathy (NCC) with severe developmental defects in the ventricular wall. With this disease-relevant in vivo model, we further unveiled previously-unrecognized roles of Tet-mediated DNA hydroxymethylation in regulating chromatin accessibility to facilitate the genomic recruitment of one key transcription factor, Ying-Yang 1 (YY1). Excitingly, our study uncovered a crucial role of Tet/5hmC in modulating long-range chromatin interactions to coordinate higher-order chromatin organization during embryonic heart development.

## Results

### Dynamic 5mC and 5hmC changes during heart development.
To evaluate DNA methylation dynamics during mammalian heart development, we performed whole-genome bisulfite sequencing (WGBS; for 5mC profiling) and CMS-IP-seq (for 5hmC profiling) in both human and mouse embryonic heart tissues (Supplementary Table 1). For human heart tissues, we analyzed DNA methylation and hydroxymethylation at the Carnegie Stage (CS) 13 and 14, which are analogous to embryonic day 9.5 (E9.5) to E10.5 of the murine heart developmental stages[24]. For mouse hearts, we analyzed DNA methylation dynamics using WGBS data available in ENCODE[25] at different embryonic developmental stages (E11.5, 12.5, 13.5, 14.5, 15.5, 16.5, P0). We also performed CMS-IP-seq using mouse embryonic cardiac tissues to compare 5hmC levels at the E12.5 stage.

We first comprehensively analyzed the DNA methylation dynamics in mouse hearts since the ENCODE data covered most of the key cardiac developmental stages. Although the global DNA methylation levels remained stable during embryonic development, ranging from 0.736 to 0.755 across all the stages (Fig. 1a), we were still able to locate 21,467 differentially methylated regions (DMRs, defined as >20% methylation change, FDR ≤ 0.05) that covered 105,710 CpG sites from E11.5 to P0 (Fig. 1b, Supplementary Fig. 1A), revealing dynamic changes in focal rather than global DNA methylation during cardiac development. Notably, very few DMRs (1%) were commonly shared among the analyzed developmental stages (Supplementary Fig. 1B), and the majority of DMRs (~ 99%) were identified at different genomic regions for each developmental stage. This finding suggests that the observed focal DNA methylation changes are stage-specific, rather than occurring at the same genomic regions, during embryonic heart development. DNA Genomic Regions Enrichment of Annotations Tool (GREAT)[26] analysis on all the identified DMRs further revealed that these regions are enriched at the cis-regulatory elements of genes essential for embryonic development and cardiac function, such as Bmp10 and Tnnt2 (Fig. 1c, d). Among all identified DMRs between adjacent developmental stages, approximately 66% (14,155 out of 21,476 DMRs) exhibited a reduction in DNA methylation when progressing to the next developmental stage (defined as hypoDMRs; Fig. 1b). Notably, more than 80% of DMRs were classified as hypoDMRs during the E12.5-to-E13.5 or E16.5-to-P0 transitions (Fig. 1b), suggesting a pronounced reduction in local DNA methylation possibly through DNA demethylation in these development stages.

Since Tet-mediated DNA methylation oxidation, particularly DNA hydroxymethylation, is a key intermediate step for DNA demethylation during development, we next measured the global changes of 5hmC by a dot-blot assay[27,28] in murine heart tissues collected at different developmental stages (E12.5, 14.5, 16.5, 18.5 and P0; Fig. 1e)[13,29]. We observed a gradual increase of 5hmC during heart development (Fig. 1e), suggesting that Tet-mediated DNA hydroxymethylation regulates DNA methylation dynamics during heart development. Real-time quantitative PCR (qPCR) also unveiled dynamic changes in Tet expression in cardiac tissues isolated at these developmental stages (Supplementary Fig. 1C). Immunohistochemistry (IHC) staining in E12.5 heart tissues revealed strong signals for 5mC, 5hmC, 5fC, and to a lesser extent, for 5caC (Supplementary Fig. 1D). In parallel, IHC analysis of Tet1 and Tet2 at the same developmental stage (E12.5) confirmed strong Tet1 expression in myocardium; while Tet2 expression was abundantly detected in all three layers of heart wall, including myocardium, epicardium and endocardium (Supplementary Fig. 1E). Similarly, we observed strong 5hmC signals in human embryonic heart tissues at CS12 and 14 (Supplementary Fig. 1F), as well as strong IHC signals for TET1 and TET2 staining across heart tissues (Supplementary Fig. 1G). Real-time qPCR analysis revealed dynamic changes in the expression of TET and DNMT family members in the early stages of human heart development (CS11-14) (Supplementary Fig. 1H). The lack of Tet3 IHC analysis is due to the unavailability of a reliable Tet3 antibody tailored for IHC analysis.

To directly investigate 5hmC distribution, we performed CMS-IP-seq for genome-wide 5hmC profiling on isolated E12.5 mouse embryonic hearts. We observed strong 5hmC enrichment at identified DMRs (Fig. 1b) during heart development, with higher enrichment of 5hmC at hypoDMRs than at hyperDMRs

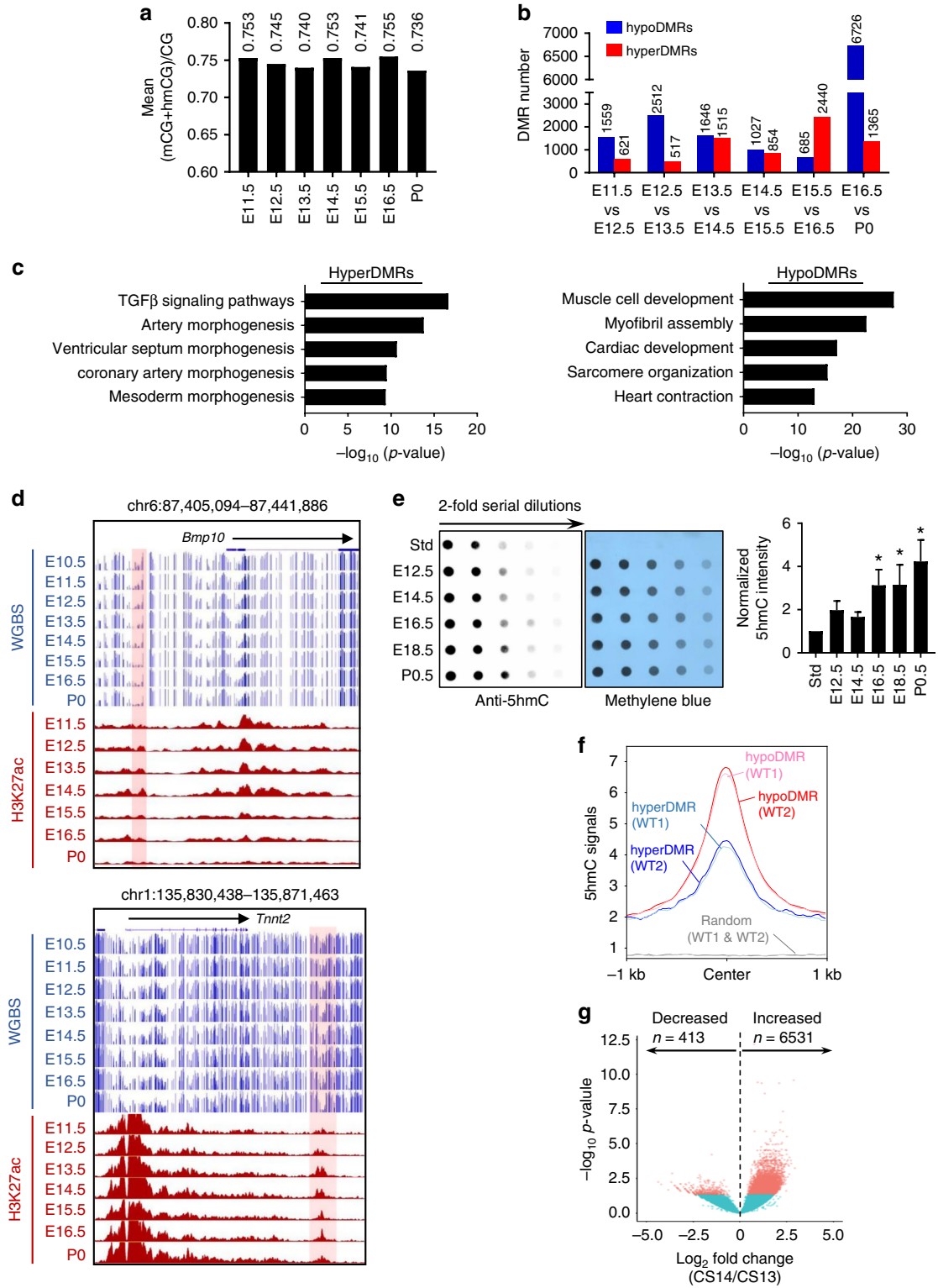

(hyperDMR defined as DMRs with significant increase of 5mC signal when transitioning into the next developmental stage; Fig. 1f). We further analyzed DNA methylation and 5hmC enrichment in human embryonic heart tissues at CS13 and 14. We noticed a pronounced increase of 5hmC during the CS13-to-CS14 transition (Fig. 1g, Supplementary Fig. 2A), coinciding with the strong induction of *TET2* and *TET3* expression at CS14 (Supplementary Fig. 1F). GREAT analysis implied that the genomic regions displaying significant differential DNA

hydroxymethylation (designated DHMRs for differential hydroxymethylated regions) between the CS13 and CS14 stages are closely associated with human embryonic and cardiac development (Supplementary Fig. 2B). For example, we observed an increase of 5hmC at proximal regions to *NOTCH* and *NKX2.5*, two genes that are essential for normal heart development (Supplementary Fig. 2C). Consistent with murine DNA methylation analysis, we did not detect significant global DNA methylation changes between the CS13 to CS14 developmental

**Fig. 1** Dynamic changes of DNA methylation and hydroxymethylation during embryonic heart development in mouse and human. **a** Quantification of global average DNA methylation levels (quantified as mean (mCG + hmCG)/CG) during mouse heart development (E11.5 to P0) based on WGBS data from ENCODE. **b** Numbers of differentially methylated regions (DMRs) that show increased (defined as hyperDMRs) or reduced DNA methylation (hypoDMRs) when mouse embryonic hearts progress into the next developmental stage. **c** GREAT analysis on hyperDMRs and hypoDMRs shown in Fig. 1b revealed representative terms associated with embryonic development and cardiac functions. Corrected binomial raw p-value were calculated. **d** Representative genome browser views illustrating the focal DNA methylation (WGBS; blue) and H3K27ac (red) dynamics at genomic regions surrounding *Bmp10* and *Tnnt2* during cardiac development in mouse embryos (E10.5 to P0). **e** (Left) Global 5hmC levels in mouse heart tissues collected at five embryonic stages (E12.5, 14.5, 16.5, 18.5, and P0) measured by the dot-blot assay. Methylene blue was used as loading control for the total DNA input. (Right) Quantification of dot-blot assay. Data were shown as mean ± S.D; $n = 3$ independent experiments. *$p < 0.05$, compared with E12.5 (two-tailed Student's $t$-test were used). **f** 5hmC enrichment signals of control E12.5 heart tissues within hypoDMRs (two repeated samples; red and pink) and hyperDMRs (two repeated samples; blue and cyan) identified from Fig. 1b. Random genomic regions (gray) were used as control. **g** The Volcano plot of differentially enriched 5hmC regions (DHMRs; CS13 *vs* CS14) in human embryonic hearts

stages of human heart (Supplementary Fig. 2D). We detected 324 and 182 hypo- and hyper-DMRs, respectively, in the CS13-to-CS14 transition (Supplementary Fig. 2E), suggesting focal DNA methylation dynamics during human heart development. Furthermore, most genomic regions displaying increased DNA hydroxymethylation (hyperDHMRs) exhibited reduction in DNA methylation (Supplementary Fig. 2F), suggesting 5hmC-mediated focal DNA demethylation during the CS13-to-CS14 transition. Taken together, our epigenomic analyses validated that focal 5mC changes, arising from altered DNA hydroxymethylation, are closely associated with murine and human heart development.

**Cardiac Tet2/3 loss causes non-compaction cardiomyopathy.** To elucidate the function of Tet-mediated DNA methylation oxidation during heart development, we generated a cardiac tissue-specific *Tet*-deficient mouse model. Earlier studies have shown that *Tet1* and *Tet2* individual knockout mice display no overt cardiac phenotypes[30–34]. Meanwhile, germline knockout of *Tet3* resulted in embryonic lethality, preventing systematic studies on cardiac development[3]. To circumvent these limitations, we first crossed mice bearing a conditional *Tet3*[flox/flox] allele[1] with the cardiomyocyte (CM) progenitor driver line, Nkx2.5-Cre[35], to yield cardiac-specific deletion of *Tet3*. Next, we crossed Tet3[flox/flox];Nkx2.5-Cre mice with Tet2KO mice to disrupt both *Tet2* and *Tet3* genes in CM progenitors (abbreviated as Tet2/3-DKO; Supplementary Fig. 3A, B). While the heterozygous mice are viable and showed no obvious cardiac phenotypes, homozygous mice were embryonically lethal (Fig. 2a). We collected embryos at developmental stages starting from E12.5 for further analyses. No appreciable morphological abnormalities were observed in Tet2/3-DKO embryos at E12.5 and E15.5, although some Tet2/3-DKO embryos had evidence of hemorrhage (Supplementary Fig. 3C). After histological analysis on embryonic hearts collected at E12.5, 13.5, 14.5 and 15.5, we found that Tet2/3-DKO embryos displayed severe cardiac developmental defects, including ventricular septal defect (VSD) and double outlet right ventricle (DORV) (Supplementary Fig. 3D). Tet2/3-DKO hearts showed abnormal ventricular chamber development starting from E13.5 (Fig. 2b). Specifically, Tet2/3-DKO hearts had ventricular non-compaction cardiomyopathy (NCC) phenotype, as evidenced by significantly reduced ventricular wall thickness and increased trabecular areas compared to controls (Fig. 2b, Supplementary Fig. 3D, E).

Next, we performed real-time qPCR to examine the expression of *Nppa* and *Hey2*, previously known to be implicated in ventricular NCC, in Tet2/3-DKO hearts (Fig. 2c). We detected a significant decrease of *Nppa* and *Hey2* expression in Tet2/3-DKO, thus validating the NCC phenotype at the molecular level. Because ventricular NCC has been shown to arise from defects in CM proliferation and/or increased cellular apoptosis[36], we further performed immunofluorescent (IF) staining with cellular proliferation and apoptotic markers in WT and Tet2/3-DKO E12.5

hearts. We observed a significant reduction in the staining signals for the cellular proliferation marker Ki67 (Fig. 2d). With regard to cleaved caspase-3 as apoptotic marker, we failed to detect meaningful signals in E12.5 control and Tet2/3-DKO heart tissue (Supplementary Fig. 3G). Together, these results clearly established the physiological roles of Tet2 and Tet3 in mediating ventricular chamber development.

**Tet2/3 regulate cardiac-specific transcription.** We next performed transcriptomic profiling, with RNA-seq, in cardiac tissues collected from E12.5 and E15.5 embryos (Supplementary Table 1, Supplementary Fig. 4A). We identified a total of 2,101 differentially expressed genes (DEGs), with 1,268 down-regulated and 833 up-regulated, respectively, in Tet2/3 DKO samples collected at E12.5 embryos compared with controls (Fig. 3a). In parallel, we identified 374 up-regulated and 440 down-regulated genes in E15.5 embryonic hearts (Fig. 3a). Gene ontology (GO) analysis on these DEGs revealed the involvement of key signaling pathways (e.g., Notch and Bmp related signaling) that are known to be crucial for ventricular chamber development (Fig. 3b, Supplementary Fig. 4B-D)[37,38]. Furthermore, key genes involved in CM development (such as *Nppa*, *Tnnt2*, and *Myh6*) showed altered expression in the Tet2/3-DKO group (Supplementary Fig. 4B–D). These unbiased transcriptomic and bioinformatic analysis data provided further evidence to support a critical role of Tet2/3 in embryonic heart development.

To further examine the function of these Tet2/3-regulated DEGs identified from RNA-seq analysis, we sorted DEGs implicated in cardiac development based on GO analysis (Fig. 3c, left). Then we analyzed the expression levels of these genes using ENCODE RNA-seq data collected from mouse embryonic hearts at different developmental stages (E10.5 to P0; Fig. 3c, right). Interestingly, we found that the expression levels of these genes underwent gradual changes during heart development, suggesting that these Tet2/3-regulated genes are tightly and temporally controlled at different embryonic stages. For example, the expression levels of *Myl2*, *Tnnt2* and *Nppa* gradually increased during normal heart development based on ENCODE RNA-seq data; however, the expression of these genes were significantly decreased in Tet2/3-DKO heart tissue (Supplementary Fig. 4E), suggesting that deletion of Tet proteins disrupts the precise transcriptional regulation of these key genes to impair cardiac development.

**scRNA-seq analysis in cardiac-specific Tet2/3 deficient mice.** The above RNA-seq analyses were performed in bulk embryonic heart tissues collected at E12.5 and E15.5 that contain multiple cell types, including myocardium, epicardium, endocardium, fibroblasts and other non-heart tissue cells (e.g., hematopoietic cells). To avoid potentially biased results due to changes in cell types upon Tet2/3 knockout, we carried out single-cell RNA-seq

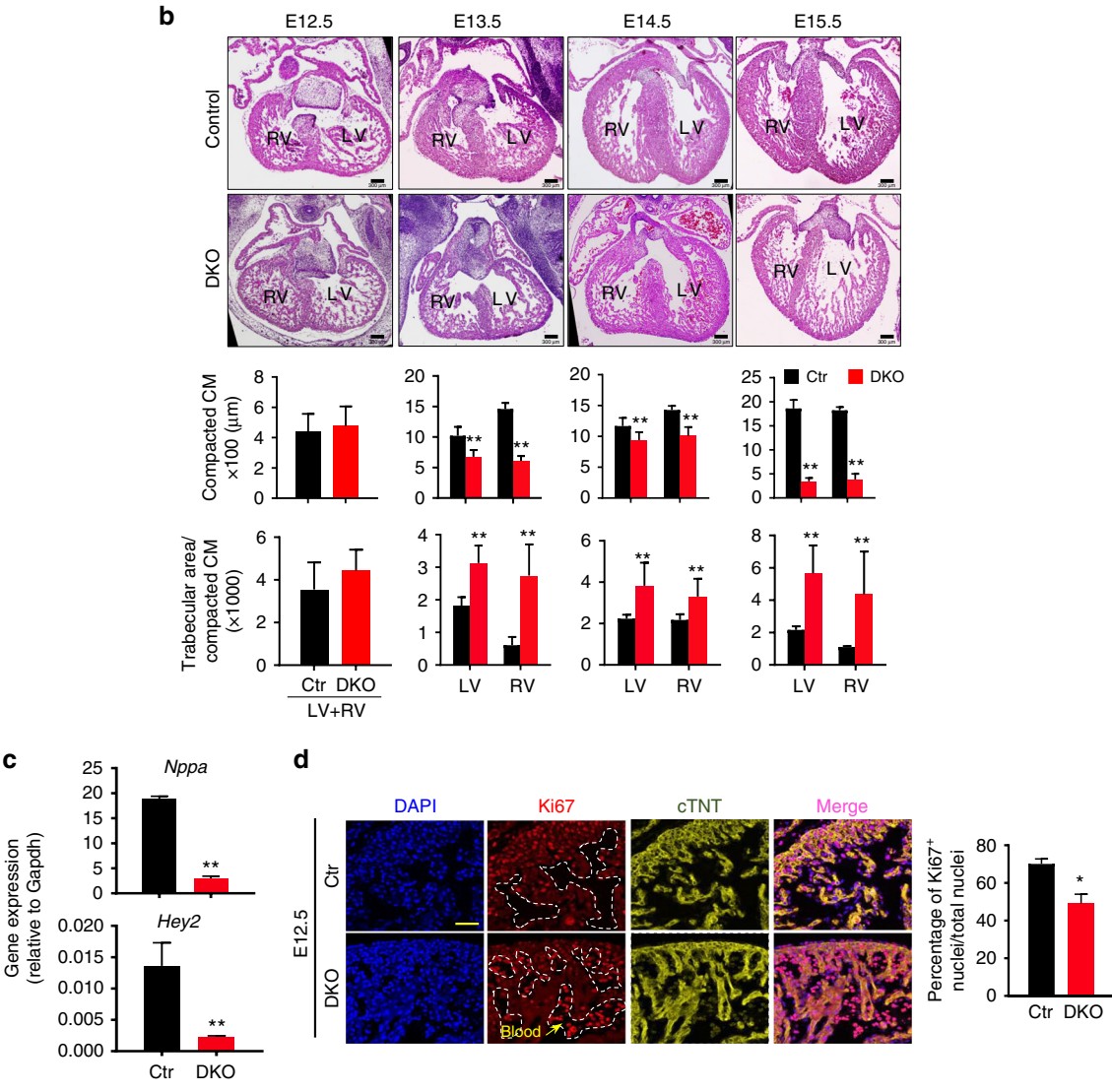

(scRNA-seq) on control and Tet2/3-DKO E12.5 and E15.5 hearts (Supplementary Table 1 and 3). Consistent with our histological analysis data (Fig. 2c), we observed minor differences in the cell types in E12.5 hearts between control and Tet2/3-DKO embryos. However, we noticed a massive reduction of the CMs (from 32.9% to 6.8%) in the Tet2/3-DKO group at E15.5 (Fig. 3d, Supplementary Fig. 4F-G). Furthermore, consistent with our real-time qPCR analysis and bulk RNA-seq data described above (Figs 2d and 3a), key genes involved in ventricular compaction,

such as *Nppa* and *Hey2*, were significantly down-regulated in E15.5 Tet2/3-DKO CMs when compared to controls (Fig. 3e). Furthermore, genes essential for cardiac development, such as *Tbx20, Ttn* and *Gja1*, were significantly down-regulated in Tet2/3-DKO E15.5 CMs (Fig. 3e).

Since the CMs is mostly affected at the E15.5 stage, we then performed cluster analysis in E12.5 and E15.5 heart tissues from both control and Tet2/3-DKO mice using the MAGIC algorithm[39]. In the CMs marked by *Ttn*, we found that Tet2/3

**Fig. 2** Cardiac-specific deletion of *Tet2* and *Tet3* resulted in developmental defects in the ventricular chamber. **a** Genotyping analysis from Tet2$^{-/-}$Tet3$^{flox/flox}$ and Tet2$^{+/-}$Tet3$^{flox/flox}$;Nkx2.5-Cre interbreedings. *$p < 0.05$, **$p < 0.01$ (chi-squared test were used). **b** (Top) Representative H&E staining images of embryonic heart tissues (×4) collected at E12.5, 13.5, 14.5, and 15.5 stages from control and Tet2/3-DKO mice. (Bottom) Quantifications were performed by using the Image J software. Data were shown as mean ± S.D; $n = 36$ sections from 3 independent experiments. **$p < 0.01$ compared to control (two-tailed Student's *t*-test were used). Scale bar: 300 μm. **c** Real-time qPCR to quantify the expression of *Nppa* and *Hey2* in control and Tet2/3-DKO embryonic heart tissues collected at E12.5. Data were shown as mean ± S.D; $n = 3$ independent experiments. **$p < 0.01$ (two-tailed Student's *t*-test were used). **d** (Left) Representative fluorescent imaging of embryonic heart tissues collected from E12.5 of control (top) or Tet2/3-DKO mice (bottom). DAPI (blue) was used for nuclear staining; Ki67 (red) was used as a proliferation marker; and cTnT (yellow) was used as the staining marker for cardiomyocytes. Cells demarcated within the white dashed lines are blood cells. (Right) Quantifications of the percentage of cardiomyocytes with positive Ki67 staining. Data were shown as mean ± S.D; $n = 4$ independent experiments (a total of 525 and 568 cardiomyocytes were quantified from control and Tet2/3-DKO embryos, respectively). *$p < 0.05$ compared to control (two-tailed Student's *t*-test were used). Scale bar: 50 μm

depletion led to substantial changes in the distribution patterns of principal components at E15.5 (Fig. 3f), but not at E12.5 (Supplementary Fig. 4H). Tet2/3-DKO CMs are clustered at *Ttn*-low expressed cells compared with control in E15.5 stage (Fig. 3f). Next, we performed further analyses on CMs based on the expression levels of two well-known cardiac chamber development-related genes *Tbx20* and *Hey2*, and found a positive correlation between the expression levels of these two genes in normal E15.5 CMs (Fig. 3g), implying that the abundancy of the expression of *Tbx20/Hey2* might be correlated with the maturation status of CMs. By contrast, E15.5 CMs collected from Tet2/3-DKO embryos were all clustered into a prominent *Tbx20*$^{low}$*Hey2*$^{low}$*Ttn*$^{low}$ population (Fig. 3g), suggesting that cardiac-specific Tet2/3 deletion might block CMs maturation during embryonic development. Together, these data indicated that Tet2 and Tet3 are essential for regulating the expression of genes that are critical for ventricular maturation.

**Global decrease of 5hmC in Tet2/3-deficient embryonic heart**. We next monitored the global changes of 5hmC in E12.5 Tet2/3-DKO heart tissues by immunofluorescent (IF) staining and the dot-blot assay (Fig. 4a, b). Both methods confirmed a substantial decrease of 5hmC in Tet2/3-DKO heart tissues when compared to the control group. The residual 5hmC signals could be ascribed to the existence of Tet1 or cell types other than CMs. To further delineate the function of 5hmC in regulating cardiac-specific gene expression, we performed CMS-IP-seq[40,41] to profile genome-wide 5hmC levels in E12.5 heart tissues (Supplementary Table 1, Supplementary Fig. 5A-B). Consistent with the IF and dot-blot results, we observed a global decrease of 5hmC in Tet2/3-DKO samples (Fig. 4c, Supplementary Fig. 5B). Among 9,559 identified DHMRs between the control and Tet2/3-DKO groups, we detected 8,846 genomic regions with reduced hydroxymethylation (defined as hypoDHMRs) and only 713 regions with increased DNA hydroxymethylation (designated hyperDHMRs) in Tet2/3-KO heart tissues (Fig. 4c). GREAT analysis revealed that hypoDHMRs were primarily enriched at distal regulatory regions of genes, many of which are known to be important for heart development (e.g., genes involved in Notch pathways) (Fig. 4d). Next, to assess the potential biological functions correlated with DHMRs, we further analyzed histone enrichment and DNA methylation levels within the identified DHMRs using the ENCODE data from E12.5 murine heart tissues. We found that DNA methylation levels at these Tet2/3-regulated hypoDHMRs were relatively low (with median DNA methylation level at 0.30) (Supplementary Fig. 5C). Furthermore, these regions were highly enriched with H3K4me1 and H3K27Ac, which are usually marked at enhancers (Fig. 4e, Supplementary Fig. 5D). By contrast, the average DNA methylation of hyperDHMRs were relatively high with a median level at 0.88 (Supplementary Fig. 5C), accompanied by moderate enrichment of H3K36me3 but no

other histone marks (Fig. 4e, Supplementary Fig. 5D). Notably, DNA methylation levels at hypoDHMRs were found to undergo larger fluctuations than hyperDMRs (Supplementary Fig. 5E) during cardiac development. In summary, our unbiased epigenomic analyses suggested that Tet2/3-mediated DNA hydroxymethylation reshapes the epigenetic status of genomic regions that are important for transcriptional regulation during heart development.

Next, we aimed to address whether Tet/5hmC loss alters the DNA methylation during cardiac development. We measured DNA methylation in Tet2/3-DKO heart tissues collected at E12.5 using whole genome-wide bisulfite sequencing (WGBS) analysis (~30× coverage of CpGs). In parallel, we compared our own Tet2/3-KO WGBS data with ENCODE WGBS data collected with E12.5 WT heart tissues. We noted a slight increase of the average DNA methylation level (mCG + hmC/CG) in the DKO group (Fig. 4f, Supplementary Fig. 5F). We next compared the DNA methylation levels at each CpG sites between the control and Tet2/3-DKO groups. We identified 13,377 and 27,880 hyper- or hypo-differentially methylated regions (DMRs) in the Tet2/3-DKO group (Supplementary Fig. 5G). GREAT analysis showed that hyper-DMRs are enriched at genes closely associated with heart function (Fig. 4g). We further analyzed DNA methylation within identified hypoDHMRs and observed that the majority of hypoDHMRs displayed increased DNA methylation (Fig. 4h, Supplementary Fig. 5H), suggesting that Tet-mediated DNA hydroxymethylation indeed mediates DNA demethylation during cardiac development. In addition, we observed a number of hypo-DMRs in Tet2/3-DKO heart tissue (Supplementary Fig. 5G) which might be due to the cross-talk between Tet proteins and Dnmt families[42]. Unlike hyper-DMRs, the function of these hypo-DMRs are not clear: they are not associated with genes involved in regulating cardiac function (Supplementary Fig. 5I) and are not co-enriched with histone modifications (Supplementary Fig. 5J). Further studies are needed to clarify the regulation and function of hypo-DMRs in the Tet2/3-DKO group.

In addition, we compared the alterations in DNA methylation or hydroxymethylation with changes in gene expression between WT and Tet2/3-DKO heart tissues collected from E12.5 embryos. A significant fraction of DEGs displayed increased DNA methylation (43.1%) and decreased hydroxymethylation (39.8%) in the Tet2/3-DKO group, respectively (Supplementary Fig. 5K). These results suggest that DNA methylation and hydroxymethylation at least partially contributed to transcriptional regulation during early heart development.

**5hmC loss perturbs YY1 binding and chromatin accessibility**. Tet-mediated DNA hydroxymethylation has been correlated with chromatin accessibility[10–12]. This prompted us to examine the genome-wide changes of chromatin accessibility by using ATAC-seq (Supplementary Table 1, Supplementary Fig. 6A, B) in Tet2/3-DKO embryonic heart. Upon Tet2/3 deletion, a total of 2816

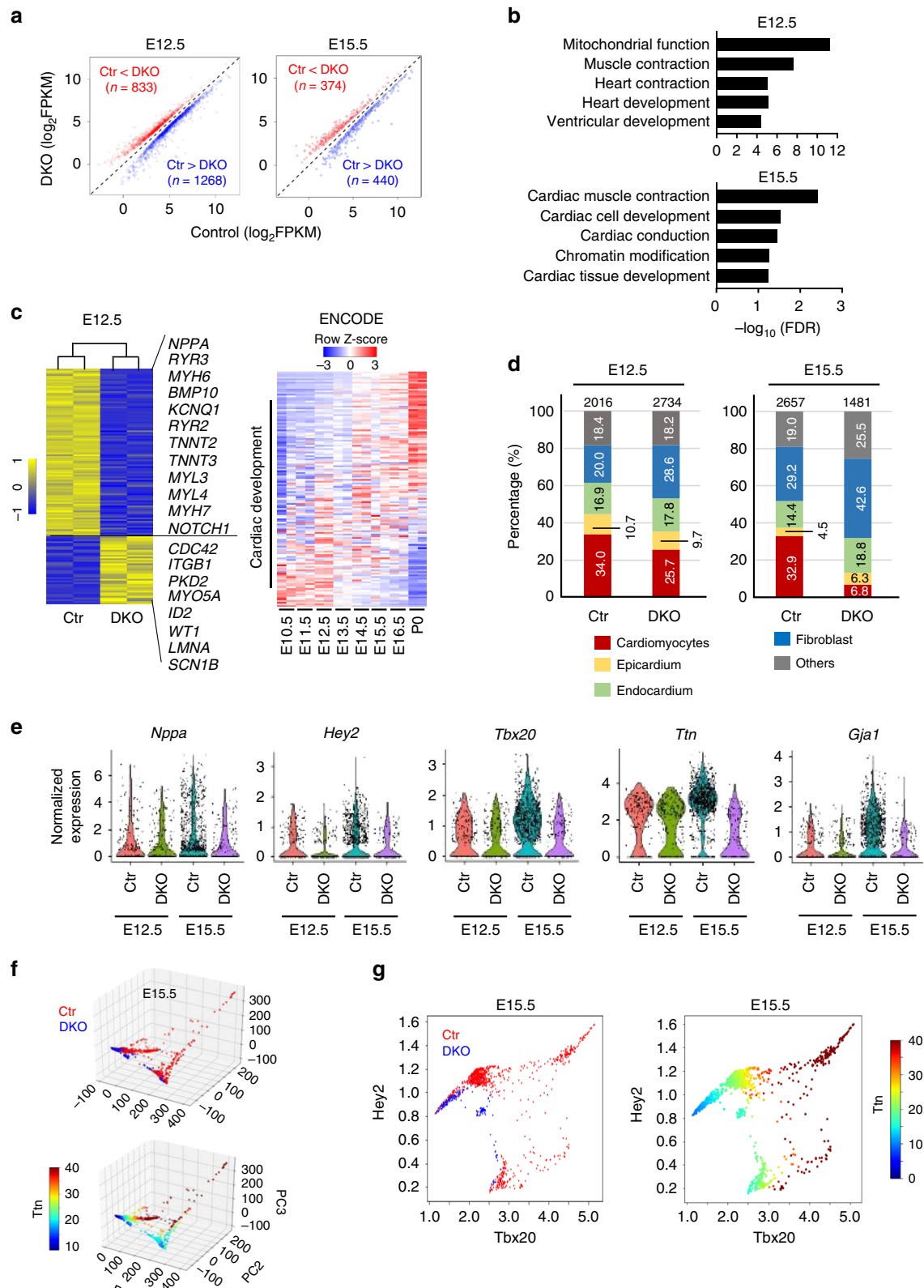

and 960 genomic regions displayed reduced and increased chromatin accessibility, respectively (Fig. 5a). Interestingly, we observed a strong positive correlation between 5hmC and chromatin accessible regions in both control and Tet2/3-DKO heart tissues (Pearson correlation coefficient of 0.85; Fig. 5b). More than 61% of ATAC-seq peaks overlapped with 5hmC-enriched

regions (Supplementary Fig. 6C). Subsequently, we selected genomic regions displaying altered 5hmC levels or chromatin accessibility in the Tet2/3-DKO group and found that 68.4% of selected regions showed simultaneous reduction in 5hmC and chromatin accessibility (Fig. 5c, Supplementary Fig. 6D). GREAT analysis revealed that these overlapping regions were mostly

**Fig. 3** Transcriptomic analyses on control and Tet2/3-DKO embryonic heart tissues. **a** Scatter plot of the RNA-seq expression data to identify differentially expressed genes (DEGs) in embryonic heart tissues between the control and Tet2/3-DKO groups at the E12.5 (left) or E15.5 (right) developmental stages. DEGs were defined as q-value < = 0.05. Red and blue dots stand for up- and down-regulated genes, respectively, in the Tet2/3-DKO group when compared to control. **b** GSEA analysis of DEGs identified between control and Tet2/3-DKO embryonic heart tissues collected at E12.5 (top) and E15.5 (bottom). Benjamini–Hochberg corrected hypergeometric *p*-value were used. **c** (Left) Heatmap presentation of the cardiac development-associated DEGs in control and Tet2/3-DKO heart tissue collected at E12.5. (Right) Heatmap presentation of expression data for the same group of cardiac development-associated DEGs in embryonic heart tissues collected at different developmental stages (E10.5 to P0). RNA-seq data were obtained from ENCODE. **d** Percentages of cell types in E12.5 and E15.5 heart tissues collected from control and Tet2/3-DKO mice using single-cell RNA-seq (scRNA-seq) analysis. Numbers listed above each bar represent the total analyzed cell numbers. **e** Violin plot showing the distribution of normalized expression levels of selected DEGs at E12.5 and E15.5 heart tissues collected from the control and Tet2/3-DKO groups. CMs were selected based on the expression of cTnT in each cell. Each dot represents the expression levels of corresponding genes in single cells. **f** The 3D PCA plots (top) and *Ttn* expression levels (bottom) of individual cardiomyocytes based on scRNA-seq data in the E15.5 control (red) and Tet2/3-DKO (blue) groups. **g** Selected *Ttn*-expressing CMs from E15.5 control (red) and Tet2/3-DKO (blue) were displayed based on the expression levels of *Hey2* and *Tbx20* (left). The expression of *Ttn* in the corresponding cells were shown in the right panel

enriched at distal regulatory regions of genes important for heart development (e.g., *Mly2*, *Tnnt2*, and *Ttn*; Supplementary Fig. 6E). Motif analysis further identified the enrichment of transcription factors (TFs) binding motifs for key cardiac development genes, such as *Mef2C*, *Gata4* and *Nkx2.5* (Supplementary Fig. 6F), within these genomic regions. We next examined the effects of decreased 5hmC and ATAC on transcription level and observed positive correlation between gene expression and 5hmC/ATAC-seq signals. For instance, we observed a strong association between reduced expression of cardiac development genes (e.g., *Nppa*, *Tnni2*, and *Bmp10*) and reduced 5hmC/ATAC-seq signals within 2 kb of transcription start sites (TSS) at corresponding genes (Fig. 5d, e, Supplementary Fig. 6G). Taken together, these data suggest that cardiac specific deletion of Tet proteins induced 5hmC loss and caused a reduction in chromatin accessibility to alter cardiac gene expression.

Based on the data described above, we hypothesized that Tet deletion reduces 5hmC and chromatin accessibility, and subsequently affects key TF binding to their genomic targets in embryonic hearts. GREAT analysis on ATAC-seq data from control and Tet2/3-DKO heart tissues pointed to YY1 as the top candidate, which showed strong enrichment in genomic regions displaying reduced chromatin accessibility in the Tet2/3-DKO group (Fig. 5f). YY1, a member of the Gli-Kruppel family of zinc finger protein, is an important transcription factor regulating early heart development[43]. To test this hypothesis, we generated Tet triple knockout mouse embryonic stem cells (Tet-TKO mESC) by using the CRISPR/Cas9-based genome editing tool as previously reported[44] (Supplementary Fig. 7A–C). Since Tet-TKO mESC has an undetectable 5hmC level (Supplementary Fig. 7B), it provides a clean system to elucidate the impact of 5hmC on TF binding to chromatin. Next, we measured the chromatin association of YY1 in WT and Tet-TKO mESCs. We observed a significant decrease in chromatin associated YY1 in Tet-TKO mESCs compared with parental WT mESCs (Fig. 6a, Supplementary Fig. 8A).

To further examine the correlation between 5hmC and YY1, we applied the CUT&RUN (C&R) method[45] to enrich YY1 genomic binding regions in WT and Tet-TKO mESCs. We first compared YY1 enriched regions obtained from the C&R method with published YY1 ChIP-seq data in mESCs[46] and observed a similar pattern between these two datasets (Supplementary Table 1, Supplementary Fig. 8B, C), revealing the robustness of the YY1 C&R data. Then we examined the correlation between YY1 and 5hmC in parental WT mESCs. We observed a strong 5hmC enrichment with concomitant depletion of 5mC signals at YY1 enriched regions (Fig. 6b). About 16% of YY1-enriched peaks (n = 10,450) overlapped with 5hmC enriched regions (Fig. 6c). To further examine the impact of Tet/5hmC loss on

YY1 genomic distribution, we compared YY1 enriched regions between the WT and Tet-TKO groups and noted that 73% of YY1 peaks showed less enrichment in Tet-TKO mESCs (Fig. 6d, e, Supplementary Fig. 8D-E), particularly at genomic regions that displayed 5hmC reduction upon Tet deletion (Supplementary Fig. 8F). We also observed increased DNA methylation within YY1 binding regions in Tet-TKO mESCs (Fig. 6f). To further validate this in a physiologically-relevant context, we performed YY1 C&R in heart tissues collected from E12.5 control and Tet2/3-DKO mice. Due to the limited cell numbers in the tissue, we identified slightly lower numbers (n = 11,469) of high-confident YY1-enriched regions when compared with the YY1 peak numbers in mESCs (n = 15,055). Using these high confident YY1 enriched peaks, we found a strong reduction of YY1 enrichment in Tet2/3-DKO heart tissues (Fig. 6g, Supplementary Fig. 8G), which is consistent with results made from the mESC study (Fig. 6d, e). Likewise, we observed increased DNA methylation and reduced 5hmC enrichment within YY1-enriched regions in the Tet2/3-DKO group (Supplementary Fig. 8H, I). To further examine the impact of chromatin accessibility on YY1 binding, we compared YY1 enriched regions with ATAC-seq signals in both the control and Tet2/3-DKO groups. We noted a strong reduction of YY1 enrichment at genomic regions with reduced chromatin accessibility measured by ATAC-seq (Fig. 6h). These data further confirmed that Tet and 5hmC regulate chromatin accessibility to facilitate the binding of proper TFs, such as YY1, to their targets.

**5hmC loss disrupts higher-order chromatin structures.** YY1 is known to regulate chromatin higher-order structures by controlling promoter-enhancer looping[47]. In addition, together with others, we have reported that 5hmC is enriched at euchromatin[11–13]. Chromatin is known to be spatially categorized into two types of large compartments, A and B, that exhibit either open chromatin domain (A) or closed chromatin domain (B)[48]. To examine whether Tet mediated DNA hydroxymethylation is associated with the organization of chromatin higher-order structures, we performed HiChIP experiment in control and Tet2/3-DKO E12.5 heart tissues using an anti-smc1 antibody[49] (Supplementary Table 1, Supplementary Fig. 9A, B). Interestingly, we observed a strong enrichment of 5hmC in compartment A, but not in compartment B (Fig. 7a, Supplementary Fig. 9C), suggesting that 5hmC tends to mark transcription active regions. Then we compared three-dimensional chromatin interaction patterns between control and Tet2/3-DKO heart tissues. In general, the compartment organization between control and Tet2/3-DKO hearts showed very similar patterns (R = 0.93; Fig. 7b). However, we observed 1424 bins (50 kb resolution) switching

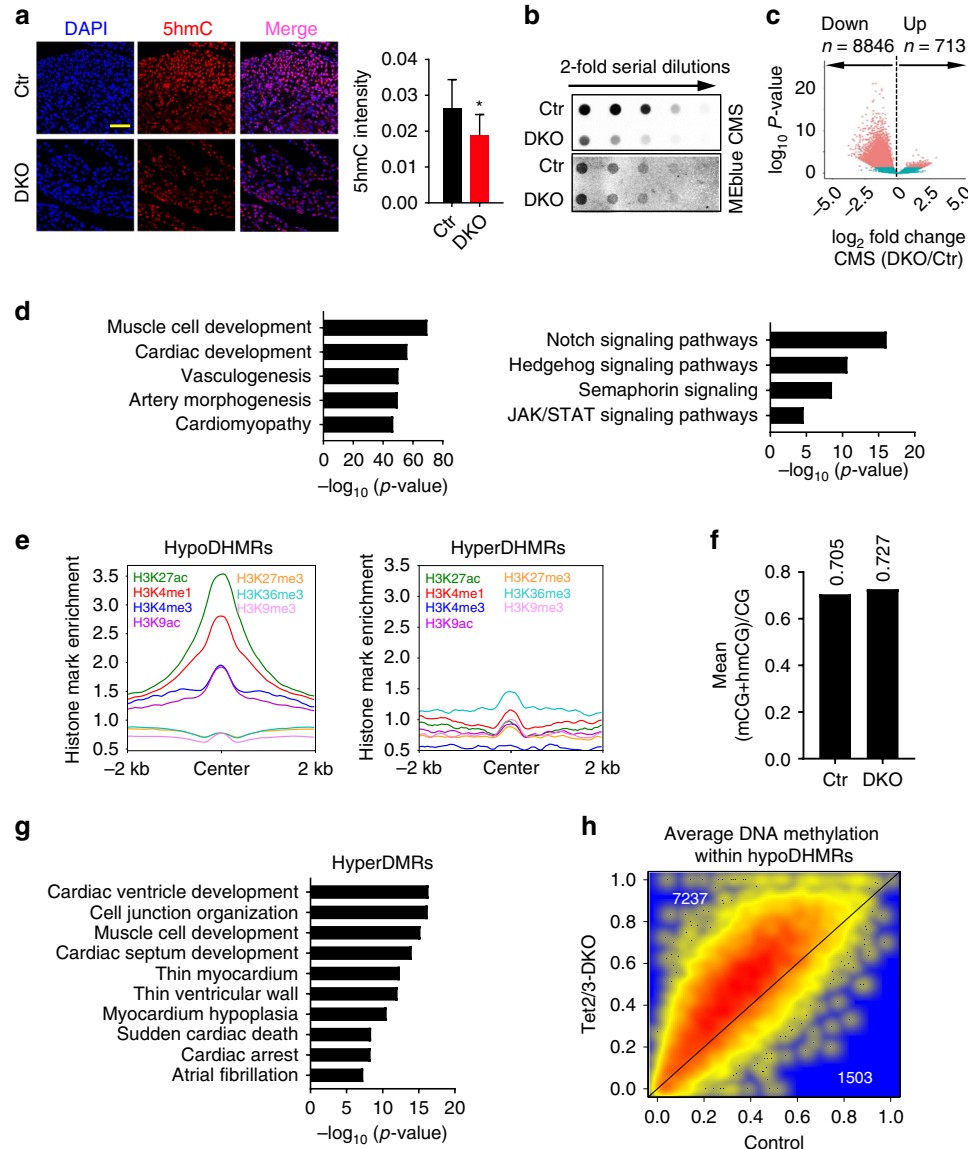

**Fig. 4** Tet2/3 deletion in embryonic heart resulted impaired 5hmC but not 5mC. **a** (Left) Representative IF staining images for control and Tet2/3-DKO murine heart tissues collected at E12.5. Blue, DAPI; Red, 5hmC. (Right) Quantification of 5hmC levels in control and Tet2/3-DKO heart tissues collected at E12.5. Data were shown as mean ± S.D; $n = 3$ independent experiments (a total of 1038 and 1250 cells were analyzed for control and Tet2/3-DKO, respectively). **$p < 0.01$ compared to control (two-tailed Student's $t$-test used). Scale bar: 50 μm. **b** Measurement of the global 5hmC levels in control and Tet2/3-DKO heart tissues collected at E12.5 by using the dot-blot assay. Methylene blue (MEblue; bottom) staining was used to visualize the total DNA input. **c** Volcano plot illustrating the differentially enriched 5hmC regions (DHMRs) in E12.5 heart tissues between the control and Tet2/3-DKO groups ($p$ value $< = 0.05$). The mean size of hypoDHMRs was 593 bp, covering 0.18% of the genome. **d** Representative GREAT analysis on hypoDHMRs illustrated in Fig. 4c. Corrected binomial raw p-value were calculated. **e** Normalized enrichment of the indicated histone modifications within identified hypoDHMRs (left) and hyperDHMRs (right). **f** Average DNA methylation levels in control and Tet2/3-DKO murine heart tissues collected at E12.5. **g** GREAT analyses on identified top 5000 most significant changed hyperDMRs in Tet2/3-DKO heart tissues compared with control. Corrected binomial raw p-value were calculated. **h** Scatterplot of the average DNA methylation levels within hypoDHMRs in E12.5 control and Tet2/3-DKO heart tissues

from compartment A to B upon Tet2/3 depletion in heart tissues (Fig. 7b). Next, we evaluated the expression levels of genes that fell into the A-to-B compartment switch category. We identified 250 down-regulated genes, with many of them known to be important for heart development, such as *Ttn, Cav1, Bmp5*, and *Actc1* (Fig. 7c); Go Ontology (GO) analysis showed that these genes are important for maintaining normal heart function or are closely implicated in cardiomyopathies (Fig. 7d).

Next, we calculated unique paired-ended tags (PETs) using the Fit-HiC pipeline[50] to identify the long-distance contacts in E12.5 heart tissues. We identified 475,630 and 347,816 confident contacts in control and Tet2/3-DKO heart tissues, respectively (Fig. 7e). We also observed strong positive association among PETs, 5hmC enriched regions and chromatin accessible regions (Supplementary Fig. 9D). We further moved on to examine the potential impact of altered contacts on gene transcription by counting PETs at DEGs identified between the control and Tet2/3-DKO groups. We noticed that 60% ($n = 979$) of DEGs with mapped PETs exhibited reduced long distance contacts in the Tet2/3-DKO group (Fig. 7f). For example, at the *Tbx20* and *Hey2* loci, two genes which displayed significant downregulation upon Tet deletion in scRNA-seq analysis, we detected a pronounced

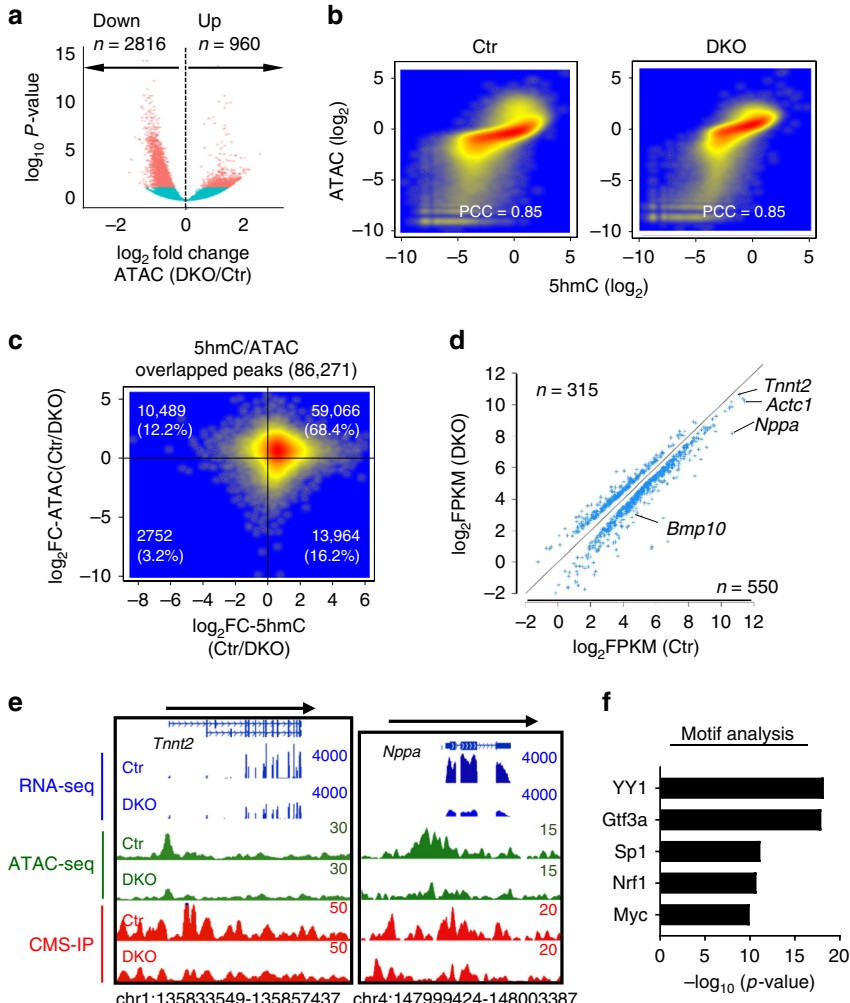

**Fig. 5** Tet2/3 deletion in murine embryonic hearts reduced chromatin accessibility. **a** Volcano plot depicting the differential ATAC-seq enriched regions between the control and Tet2/3-DKO groups (p value < = 0.05). Data were collected from E12.5 embryonic heart tissues obtained from control and Tet2/3-DKO mice. **b** Scatterplot depicting the correlation between 5hmC and ATAC-seq signals in the control (left) and Tet2/3-DKO (right) groups. Pearson correlation coefficient was calculated by correlating 5hmC signals with ATAC-seq signals obtained from E12.5 cardiac tissues. The ATAC and 5hmC signals of every 10 kb bin were calculated. **c** Scatterplot depicting the distribution of genomic regions with altered 5hmC (x-axis) and ATAC-Seq signals (y-axis) between the control and Tet2/3-DKO groups. Normalized 5hmC and ATAC signals were used in the plot. **d** Scatterplot presenting the expression of DEGs that showed decreased in 5hmC and ATAC-seq signals within 2 kb of transcription start sites (TSS). *Tnnt2*, *Actc1*, *Nppa*, and *Bmp10* were among the hits. **e** Genome browser view examples of decreased transcription (blue, RNA-seq), chromatin accessibility (green, ATAC-seq) and 5hmC (red, CMS-IP) enrichment at cardiac-specific gene loci (*Tnnt2* and *Nppa*) in Tet2/3-DKO E12.5 heart tissues compared with that in control group. **f** Motif enrichment analysis on genomic regions displaying decreased ATAC-seq signals in the Tet2/3-DKO group. Benjamini–Hochberg corrected hypergeometric p-value were used

decrease of promoter-enhancer interactions in the Tet2/3-DKO group compared to the control group (Fig. 7f, g). In parallel, we noted a concomitant reduction in 5hmC and ATAC-seq signal enrichment in the Tet2/3-DKO group within these two loci (Fig. 7g, Supplementary Fig. 9E). To further evaluate the impact of Tet/5hmC in regulating YY1 binding during the long-distance interaction, we compared Hi-ChIP signals between WT and Tet-TKO mESCs (Supplementary Fig. 9F, G). We observed a strong reduction of PETs at YY1 enriched regions in Tet-TKO mESCs compared to the parental WT ESCs (Supplementary Fig. 9G). In parallel, we carried out a functional rescue experiment by expressing the catalytic domain of Tet1 (Tet1CD) in WT or Tet-TKO mESCs. The expression of Tet1CD and subsequent increase of 5hmC in the Tet-TKO mESCs were confirmed by Western blotting (Supplementary Fig. 9H) and the 5hmC dot-blot assay (Supplementary Fig. 9I). Next, YY1 ChIP-qPCR and 4C-seq were performed at selected genomic regions to examine

YY1 binding and the chromatin looping status before and after Tet1CD re-expression (Fig. 8a, b). We selected the *Mpdu1* locus because it was among the top ranked genomic regions with decreased YY1 binding and chromatin looping upon Tet depletion. We found that Tet1CD expression in Tet-TKO mESCs significantly restored YY1 binding (Fig. 8a) and chromatin looping (Fig. 8b) at the *Mpdu1* locus, suggesting the involvement of Tet/5hmC in regulating YY1-associated long-range chromatin interactions in the genome. Together, these findings establish that Tet proteins modulate the formation of YY1-associated promote-enhancer looping and use this mechanism to regulate gene transcription.

## Discussion

Abnormal DNA methylation has been reported in multiple congenital heart diseases, suggesting that dynamic DNA

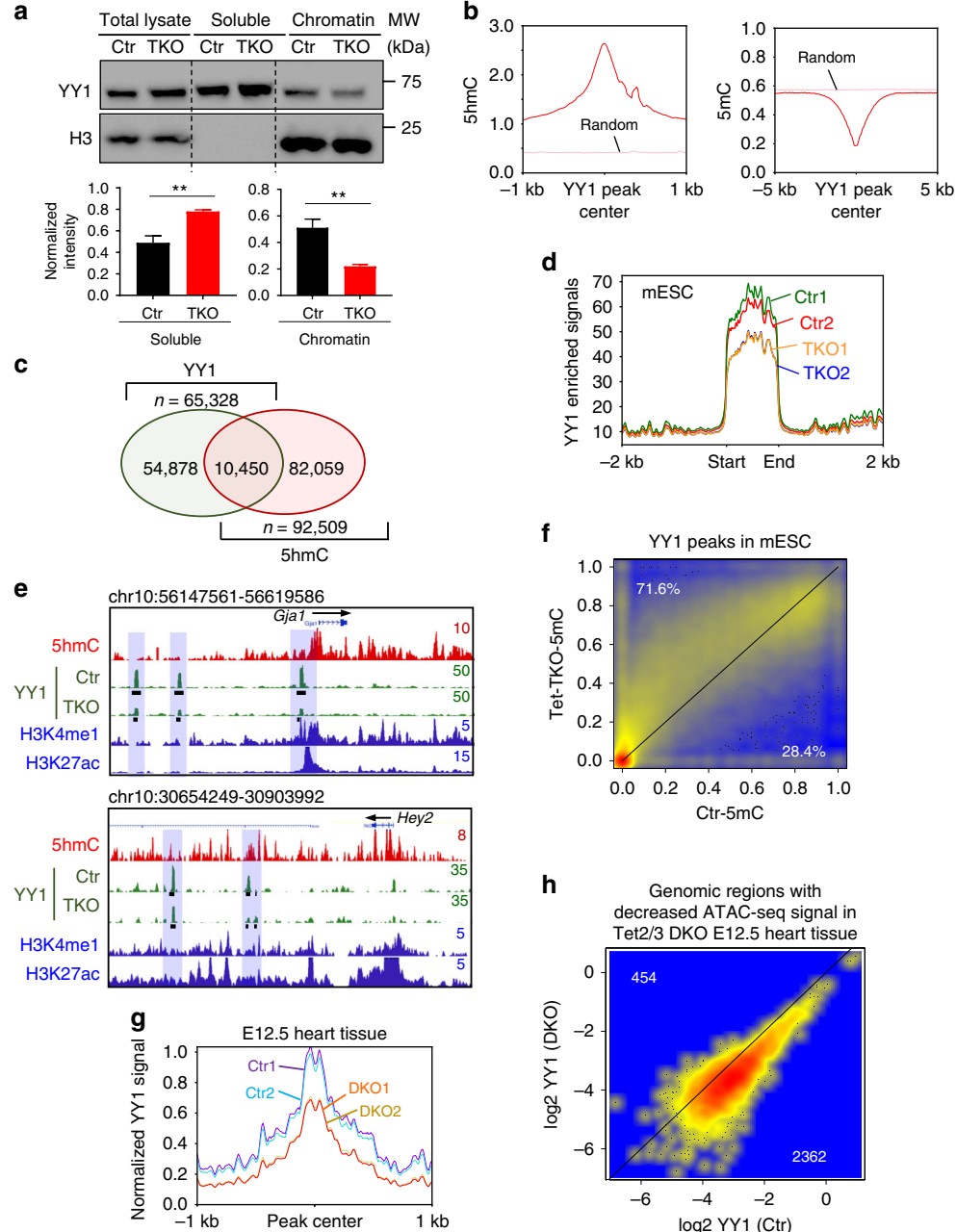

**Fig. 6** Tet2/3 deletion compromised the binding of YY1 to chromatin. **a** (Top) Representative western-blot analysis of YY1 binding to chromatin in WT and Tet-TKO mESCs. (Bottom) Quantification of the intensity of YY1 soluble and chromatin associated fractions in WT and Tet-TKO mESCs. Data were shown as mean ± S.D; $n = 3$ independent experiments. **$p < 0.01$ compared to WT (two-tailed Student's $t$-test were used). **b** Enrichment of 5hmC (left) and 5mC (right) signals within YY1-enriched peaks. Random regions were used as control. **c** Venn diagram showing overlapped peaks between YY1- and 5hmC-enriched regions in mESCs. 10,450 overlapping peaks were identified. **d** YY1 enrichment in WT (red and green) and Tet-TKO (yellow and blue) mESCs. **e** Exemplary genome-browser views of 5hmC enrichment in WT mESCs (red trace) and YY1 enrichment (green) at the *Gja1* and *Hey2* loci in WT and Tet-TKO mESCs. Representative genomic regions that showed reduction in YY1 enrichment in Tet-TKO mESCs (compared to control) were highlighted in blue. The black bars beneath the peaks indicated the called peaks. **f** Scatterplot depicting the DNA methylation levels within the YY1 enriched regions in WT ($x$-axis) and Tet-TKO ($y$-axis) mESCs. **g** YY1 enrichment in control (blue and purple) and Tet-DKO (yellow and orange) mouse embryonic heart tissues collected at the E12.5 stage. **h** Scatterplot depicting YY1 enrichment within the genomic regions that showed decreased ATAC-seq signals in control ($x$-axis) and Tet-DKO ($y$-axis) heart tissues collected at E12.5 mouse embryos

methylation is one of the important epigenetic events controlling heart development and cardiac functions[19–21]. In the current study, we have systematically analyzed DNA methylation and hydroxymethylation dynamics during embryonic heart development in both human and rodents. Our integrative studies have unveiled dynamic focal DNA methylation changes, driven by TET-mediated DNA hydroxymethylation, at genes essential for cardiac development.

The TET protein family is one of the major regulators controlling DNA methylation oxidation. The current study is primarily focused on studying Tet2 and Tet3 given their relatively higher expression in embryonic cardiac tissues. Cardiac specific

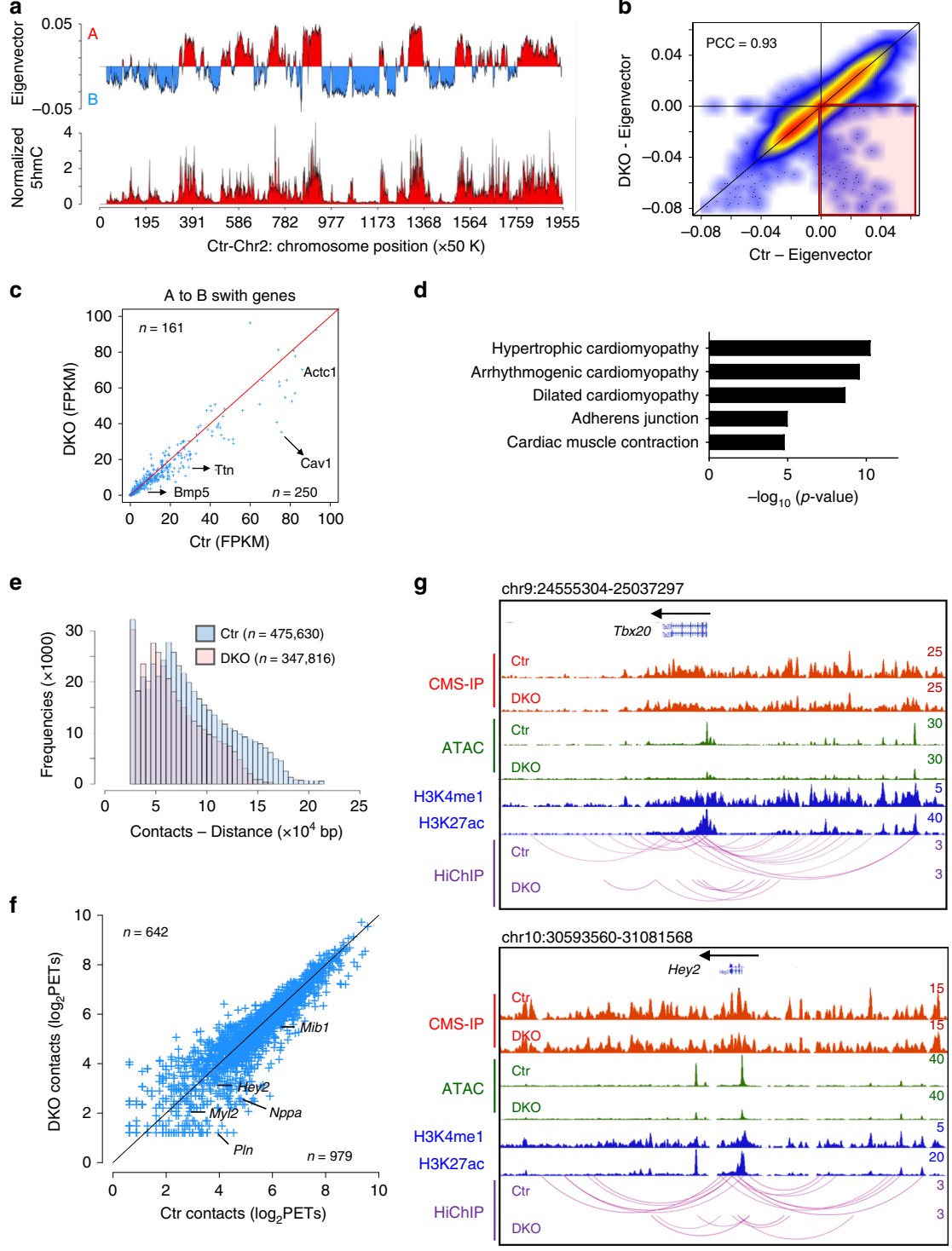

**Fig. 7** Tet-mediated DNA hydroxymethylation is associated with chromatin higher-order structures. **a** Representative histograms of compartment A and B distribution (top) calculated by Eigenvector using HiChIP and CMS-IP-seq (for 5hmC) data over chromosome 2 in E12.5 control embryonic heart tissues. **b** Scatterplot showing the distribution of Eigenvector calculated in control and Tet2/3-DKO embryonic heart tissues at E12.5. Pearson correlation coefficient was calculated. **c** Expression levels of genes that underwent A-to-B compartment switch in control and Tet2/3-DKO embryonic heart tissues (E12.5). **d** Representative GSEA analysis of genes located in genomic regions that showed A-to-B compartment switch in Tet2/3-DKO embryonic heart tissues (E12.5). **e** Histograms of PETs (distance and numbers) from HiChIP data obtained from control and Tet2/3-DKO heart tissues collected at E12.5. **f** Scatterplot showing the normalized numbers of PETs that overlapped with DEGs (regions ranging from TSS to TTS) in control and Tet2/3-DKO heart tissues collected at E12.5. **g** WashU Epigenome browser views of 5hmC, ATAC-seq, PETs (HiChIP), H3K4me1 (ENCODE) and H3K27ac (ENCODE) at the *Tbx20* and *Hey2* loci in control and Tet2/3-DKO heart tissues collected at E12.5

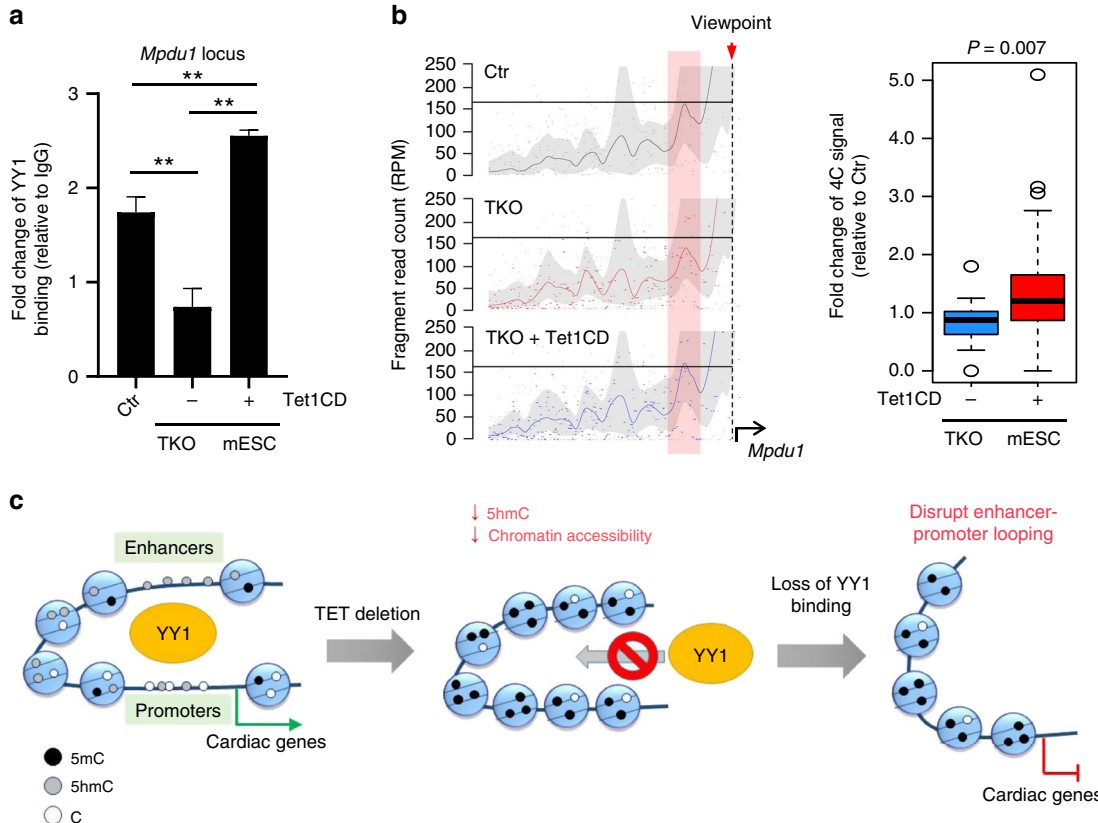

**Fig. 8** Re-expressing Tet1CD partially restores the YY1 mediated enhancer-promoter interactions. **a** ChIP-qPCR analysis of YY1 binding at the *Mpdu1* locus in TKO mESCs and TKO mESCs after expression of Tet1CD. Data were shown as mean ± S.D; $n = 2$ independent experiments. ** $p < 0.005$ (two-tailed Student's *t*-test were used). **b** Representative 4C-seq signals in WT (black), TKO (red) and TKO + Tet1CD (blue) mESCs at the *Mpdu1* locus. The line indicates the normalized 4C-seq signals (calculated by Basic4C-seq R package using two biological replicates) and the shaded areas represent the 95% confidence interval. The quantification of 4 C signals at red highlighted regions were shown in the right panel. Blue: relative fold-change of 4 C enrichment between TKO and WT (TKO/WT); Red: relative fold-change of 4 C enrichment between TKO + Tet1CD and WT (TKO + Tet1CD/WT). All the experiments were performed with biological duplicates. Kolmogorov–Smirnov test were used to calculate *p*-value. **c** Tet protein mediated DNA hydroxymethylation regulates chromatin accessibility and subsequently safeguards the binding of key cardiac development-associated transcription factors, such as YY1, to their target regions in the genome to maintain proper long distance interactions (enhancer-promoter looping). Deletion of Tet protein, with consequent 5hmC loss, could reduce chromatin accessibility to compromise YY1 binding to its genomic targets, thereby affecting long distance interactions to perturb the transcriptional networks underlying normal cardiac development

deletion of Tet2 and Tet3 using Nkx2.5-Cre resulted in ventricular non-compaction cardiomyopathy in mice, clearly attesting to the indispensable roles of these two epigenetic modifiers in normal heart development. At the molecular level, deletion of Tet2/3 prominently impaired DNA methylation/hydroxymethylation in heart tissues and altered the transcription of cardiac development associated pathways, such as Notch signaling. Mechanistically, we propose that changes in chromatin accessibility, attributed to compromised DNA hydroxymethylation, sabotage the binding of key cardiac development-associated TFs, as exemplified by YY1 (Fig. 8c), to their targets across the genome. YY1 is a key mediator of embryonic heart development by facilitating GATA4 mediated transcriptional activation and promoting cardiac progenitor cell commitment[43,51]. Interestingly, cardiac-specific YY1 knockout mice (YY1^f/f-Nkx2.5Cre) displayed very similar phenotypes as Tet2/3-DKO mice generated in the current study, with both in vivo models showing reduced embryo survival at E13.5 and decreased proliferation of CMs (with similar non-compaction cardiomyopathy manifestations)[52]. These findings strongly suggest that Tet proteins and YY1 might converge to regulate the similar transcription regulatory pathways during embryonic cardiac development. Although we presented evidence to support the notion that Tet deletion results in

reduced chromatin binding of YY1 because of altered chromatin accessibility, we cannot rule out the possibility that the binding of YY1 to the genome is directly dependent on Tet and /or DNA hydroxymethylation modifications. Further follow-on studies are needed to clarify this point.

Although we observed a positive correlation between YY1 binding and Tet-mediated DNA hydroxymethylation, a substantial number of 5hmC- or YY1-enriched regions do not overlap with each other, indicating that 5hmC might not be the only factor to regulate YY1 genomic binding. YY1 has been reported to interact with several other epigenetic regulators, such as HDACs, p300 and INO80, to enhance or repress gene transcription[53]. In parallel, Tet and 5hmC might further be implicated in other transcriptional regulatory pathways during embryonic development. For example, Tet-mediated DNA demethylation has been reported to control the lefty-nodal signaling during mouse gastrulation[2] and enhancer activities in the vertebrate phylotypic period[11]. Moreover, Tet1 and Tet3 deficiency has been shown to promote transcription variations during embryogenesis[1]. Tet1 further regulates the activity of JMJD8 to suppress epiblast target genes in post-implantation mouse embryos[54]. A complete picture of the context-dependent YY1/TET/5hmC genomic distribution and

transcriptional regulation during early embryogenesis is yet to be fully established.

In summary, our integrative genomic and epigenomic analyses have yielded a complete atlas of DNA methylomes and hydroxymethylomes representative of key developmental stages of embryonic hearts in both humans and rodents. The epigenetic landscapes depicted in our study can serve as a useful blueprint and starting point towards the full comprehension of cardiac epigenetics during early embryogenesis. Our study has uncovered previously unrecognized roles of Tet and 5hmC in gene regulation by modulating transcription factor binding and long-distance interactions at cardiac-specific genomic loci. In addition to Tet2 and Tet3, we also observed dynamic changes of Tet1 expression during heart development in both human and rodents. We cannot rule out the possibility that Tet1 is also an important contributor to embryonic heart development. Nonetheless, no heart-related phenotypes have thus far been reported upon Tet1 deletion in transgenic mice, probably due to the redundant functions of Tet homologs (Tet2 and Tet3). Further studies on a cardiac-specific Tet triple knockout mouse model might address the additional function of Tet1 during embryonic heart development. Although the current study exclusively focused on dissecting cell-autonomous mechanisms on how Tet protein might regulate the transcription of key genes involved in early cardiomyocytes development, we cannot rule out the possibility of cell non-autonomous mechanisms (such as how Tet deficiency in epicardium affect cardiomyocytes development) underlying this phenotype, which will be pursued in the follow-on studies.

## Methods

**Animal models**. Animal studies were approved by the Institutional Animal Care Use Committee (IACUC) of the Institute of Biosciences and Technology, Texas A&M University. Most mouse strains bear a C57BL/6 genetic background unless otherwise noted. Tet2[−/−33], Tet3f/f[1] and Nkx2.5-Cre (The Jackson Laboratory 024637)[55] mouse strains were reported previously. Timed pregnancies were applied and the day on which a plug was found was defined as E0.5. Mice tails were cut and boiled in 50 mM NaOH for 1 h and then neutralized in 10 mM Tris–HCL at pH7.4. PCR was carried out using the EmeraldAmp GT PCR Master Mix (TaKaRa) according to the manual. Genotyping primers are listed in Supplementary Table 2. The uncropped and unprocessed scans of the blots are available in source data file.

**Antibodies**. For IHC: Tet1 antibody was kindly provided by Dr. Leonhardt Heinrich[56] (1:100). Anti-Tet2 (Abcam ab124297, 1:100), anti-5mC (Millipore MABE146, 1:1000), anti-5hmC (Active Motif 39769, 1: 40,000), anti-5fC (Active Motif 61223, 1:2000), and anti-5caC (Active Motif 61225, 1:1000) were purchased from commercial sources.

For IF: anti-Ki67 (Abcam ab16667, 1:100), anti-cleaved caspase-3 (Cell Signaling Technologies 9661 s, 1:50); Alexa Fluor 568 goat anti-rabbit (Thermo Fisher Scientific A-11011, 1:1000), Alexa Fluor 647 goat anti-mouse (Thermo Fisher Scientific A-21235, 1:1,000)

For CUT&RUN experiments: anti-YY1 (Santa Cruz sc-7341, 1:100), anti-H3K27ac (Abcam ab4729, 1:100), Rabbit anti-mouse (Abcam ab6728, 1:100), anti-cTNT (Thermal Fisher Scientific 13-11, 1:100) were purchased.

For HiChIP: anti-Smc1 (A300-055A, Bethyl Laboratories, Inc. 2 μg/reaction) was used.

For Western Blotting: anti-YY1 (Santa Cruz sc-7341, 1:1000), and anti-H3 (abcam ab1791, 1:2000) antibodies were used.

**Human Samples**. Human embryos collecting protocol was approved by the Ethical Internal Review Board of the Xinhua Hospital, Shanghai, China. Human embryos were collected from pregnant mothers who performed clinical drug abortion at the Department of Obstetrics and Gynecology in Xinhua Hospital. Consent forms were signed. The embryonic stages of the embryos were measured by using a standard protocol reported previously [57].

**Histological analyses**. All mouse embryos were dissected in phosphates buffered saline (PBS). Embryos pictures were taken using a Nikon SMZ800N dissecting microscopy. For histological characterization, all embryos were fixed overnight in 4% PFA, then dehydrated with graded ethanol and embedded in paraffin. Sections were cut at the thickness of 7 μm. Slides were dried at 37 °C overnight and then stained with hematoxylin-eosin (H&E), as previously described[58,59]. Stained sections were imaged using a Nikon Eclipse Ci microscopy.

ImageJ was used for the measurement and quantification of histological data. The quantification method was described previously[38]. Briefly, the ventricle was divided into the apex and the basal regions and several measurements of the compact myocardium thickness were taken for each region. The average thickness was calculated. The trabecular area was measured by dividing the surface occupied by the trabeculae in the ventricle. The ratio of the trabecular area and compact myocardium thickness was used as an indicator for the size of the trabecular mesh.

**IHC and IF staining**. Immunohistochemistry (IHC) and immunofluorescence (IF) staining were performed, as previously described[58,59]. Briefly, tissue sections were dewaxed in xylene twice for 5 min each and rehydrated in a graded series of ethanol (100 to 70%). The antigens were retrieved by boiling sections in 10 mM citrate buffer (Vector Laboratories) for 20 min. For DNA modifications staining, sections were treated with 2 N HCl for 30 min to expose the epitopes and then neutralized in 100 mM Tris–HCl (pH 8.5) for 10 min. IHC was using the ImmPRES HRP Reagent Kit (Vector Laboratories) to perform the blocking and antibody incubation, and then developed by using the DAB peroxidase substrate kit (Vector Laboratories). IHC stained sections were imaged by a Nikon Eclipse Ci microscopy. For IF, 10% normal goat serum (Thermo Fisher Scientific) was used to block the unspecific antigens. After primary and 2nd antibodies incubation, 0.5 μg/ml DAPI (Thermo Fischer Scientific, D1306) was used to co-stain the nuclei. IF stained sections were imaged using a Nikon A1 confocal microscope.

**mESC culture and the generation Tet-TKO mESCs**. mESCs (E14) were cultured on MEFs in Knock-out Dulbecco's Modified Eagle's Medium (Gibco), supplemented with 15% fetal bovine serum (Omega), 0.5% penicillin-streptomycin (Gibco), 0.1 mM non-essential amino acids (Gibco), 0.1 mM 2-mercaptoetanol (Sigma), and 103 U/mL of leukemia inhibitory factor (Millipore). Tet1/2/3 triple knock-out mESCs were generated by the CRISPR/Cas9 technology as described previously[44] with slight modifications. Tet1/2/3 sgRNAs were cloned into PX458 (Addgene 48138). Three sgRNAs were transfected simultaneously into mESCs using the iMfectin DNA transfection reagent (Gendepot). mESCs transfected with the vector PX458 without sgRNA were used as control. GFP positive cells were sorted into 96-well plates by flow cytometry and individual colonies were genotyped after 7-day culture.

**RFLP analysis and amplicon sequencing**. RFLP analysis was performed as described previously[44]. sgRNAs (Supplementary Table 2) targeted to regions of Tet1/2/3 were amplified by PCR and 10 μl of products were digested with SacI, EcoRV or Xhol, respectively. Digested DNA was separated on 2% SYBR Safe (Gendepot) stained agarose gel. For amplicon sequencing, PCR products were purified using MinElute PCR purification kit (Qiagen) and libraries generated using Nextera XT DNA library prep kit (Illumina) according to their manuals. Libraries were sequenced on Illumina NextSeq 500 system using the NextSeq 500 High Output v2 Kit (Illumina, San Diego, CA) with a customized single end, single indexing (80/8-bp) format.

**Nuclear fractionation and western blot**. Nuclear fractionation in mESCs was performed as described previously[60]. Briefly, 10 million cells were washed by PBS and pelleted at 200 g for 2 min. 200 μl buffer A (10 mM HEPES, pH 7.9, 10 mM KCl, 1.5 mM MgCl₂, 0.34 M Sucrose, 10% Glycerol, 0.1% Triton X-100, 1 mM DTT, and protease inhibitor cocktail) was added to the cell pellets and incubated on ice for 8 min to remove the cytoplasm. After centrifugation at 1300 g, 4 °C, for 5 min, 100 μl Buffer N (15 mM Tris-HCl [pH 7.5], 200 mM NaCl, 60 mM KCl, 5 mM MgCl₂, 1 mM CaCl₂, 0.3% NP-40, and protease inhibitor cocktail) was added to the nuclear pellets and incubated on ice for 30 min to lyse. After centrifugation at 1700 g, 4 °C, for 5 min, the supernatant was collected and labeled as soluble fraction, and 100 μl sample loading buffer was added to the chromatin pellets for denaturing. Denatured proteins were loaded to the 4–12% gradient SDS-PAGE (GenScript). Nitrocellulose membranes (Millipore) were used for transferring after gel running. After blocking in 5% non-fat milk, the membranes were probed with the corresponding primary antibodies overnight at 4 °C, followed by incubation with a secondary antibody at room temperature for 1 h. After adding the West-Q Pico Dura ECL Solution (Gendeport), the antigen–antibody complexes were detected by the ChemiDoc Imaging system (Bio-Rad). The intensity of protein bands was measured by the Image Lab software package (Bio-Rad). The uncropped and unprocessed scans of the blots are available in source data file.

**Expression of FLAG-tagged Tet1CD**. FLAG-tagged Tet1 catalytic domain (FLAG-Tet1CD) was amplified from Fuw-dCas9-Tet1CD (Addgene #84475) using the primers listed in Supplementary Table 2, and then inserted into a lentiviral vector, 213-PRRL-CAG-NLS-sfGFP (a gift from Dr. Courtney Hodge at Baylor College of Medicine) between the restriction sites NheI (NEB R0131s) and EcoRV (NEB R0195S). The lentiviral vector encoding FLAG-Tet1CD plasmid was transfected into HEK 293 T cells, along with standard virus packaging vectors psPAX2 (Addgene #12260) and Pmd2.G (Addgene #12259). The virus-containing supernatants were collected 2 days after transfection followed by brief centrifugation (750 g at 4 °C for 10 min). The supernatant with packaged viruses were added into Tet TKO mESCs followed by centrifugation (750 g at 37 °C for 90 min). Tet1CD

expression and subsequent DNA hydroxymethylation was confirmed by Western blotting and a 5hmC dot blot assay, respectively, 2 days after lentivirus transduction.

**Nucleic acid isolation.** Total genomic DNA and RNA samples were isolated using the AllPrep DNA/RNA Mini Kit (Qiagen) according to the manufacturer's instructions. In brief, 600 μl of buffer RLT plus was added to the embryonic heart tissues or cell pellets. In total 26 G needles (BD) were used to disrupt samples. Lysate was transferred to an AllPrep DNA spin column and centrifuged at 10,000 × *g* for 30 s. The genomic DNA was captured by the column and 100 μl elution buffer was added to elute genomic DNA. Total RNA was in the flow-through and precipitated by one volume of 70% ethanol. Then the mixture was transferred to a RNeasy spin column. Purified total RNA was eventually eluted by 30 μl RNase-free water. DNA and RNA concentrations were measured by the Qubit fluorometer (Thermo Fisher Scientific).

**Dot-blot assay.** Purified genomic DNA was denatured in 0.4 M NaOH, 10 mM EDTA at 95 °C for 10 min, then neutralized with ice-cold 2 M ammonium acetate (pH 7.0). Two-fold serial dilutions of the denatured DNA samples were generated and spotted on a nitrocellulose membrane by using an assembled Bio-Dot apparatus (Bio-Rad) according to the manufacturer's instructions. A synthetic oligo-nucleotide with a known amount of 5hmC was used as standard[1]. The membrane was washed with 2xSSC buffer briefly, air-dried and vacuum-baked at 80 °C for 2 h. DNA hybridized membrane was blocked with 5% non-fat milk for 1 h at room temperature and incubated with an anti-5hmC antibody (1:3000, Active Motif) overnight at 4 °C. Next day, the membrane was incubated with a horseradish peroxidase-conjugated anti-rabbit IgG secondary antibody (1:10,000; Sigma) for 1 h at room temperature. The membrane was visualized by West-Q Pico Dura ECL Solution (GenDEPOT). The membrane was washed with 1× TBST briefly and then stained with 0.02% methylene blue in 0.3 M sodium acetate (pH 5.2) to confirm the total amounts of loaded DNA samples. The uncropped and unprocessed scans of the blots are available in source data file.

**Real-time quantitative PCR (qPCR).** Purified total RNA (10 pg to 5 μg) was reverse transcribed into cDNA with the amfiRivert cDNA Synthesis Platinum Master Mix (Gendepot). Real-time quantitative PCR was performed with a LightCycle 96 (Roche) instrument using amfiSure qGreen Q-PCR Master Mix (Gendepot). Three-step cycling program was used with 3 min 95 °C initial denaturation and 40 cycles of 10 s 95 °C denaturation, 20 s 60 °C annealing and 30 s 72 °C extension. All the primers were synthesized from Integrated DNA Technologies and listed in Supplementary Table 2.

**RNA-seq library construction and data analysis.** Poly-A tailed messenger RNA was enriched with a Poly(A)Purist™ MAG Kit (Thermo Fisher Scientific). Enriched mRNA was used for RNA-seq library preparation by using a NEBNext® Ultra™ Directional RNA Library Prep Kit (NEB) according to the manufacturer's instructions. The quality of libraries was checked by an Agilent High Sensitivity DNA kit (Agilent Technologies). The library was sequenced using an Illumina NextSeq 500 instrument (150 cycle, paired-end; Supplementary Table 1).

RNA-seq data were mapped to mm10 genome assembly using tophat-2.1.1 with default parameters. Cufflinks and cuffdiff were used to call significantly differentially expressed genes (DEGs) (q-value < = 0.05) between WT and Tet2/3-DKO groups at E12.5 and E15.5 developmental stages. In-house R scripts were used to plot the scatter plot for DEGs. DEGs functional enrichment was performed using GSEA[61]. RNA-seq data of mouse embryo hearts ranging from E10.5 to P0 were downloaded from ENCODE (https://www.encodeproject.org/). R package gplots was used to plot heatmaps for DEGs.

**scRNA-seq library preparation and data analysis.** Embryonic hearts were harvested and digested into single cells using 1 mg/ml collagenase I (Worthington). Single-cell RNA-seq libraries were generated using the Chromium Single-Cell 3′ Reagent V2 Kit (10× Genomics) according to the manufacturer's protocol. Briefly, single cell GEM was generated and barcoded in a Chromium Controller (10× Genomics). Then RNA transcripts from single cells were reverse transcribed, amplified and fragmented. Library generation was finished by incorporating the adapter and sample indices into the fragmented cDNA. Agilent Bioanalyzer 2100 (Agilent) was used to profile the sizes of the pre-amplified cDNA and the libraries. Libraries were subjected to highthroughput sequencing on a Illumina NextSeq 500 system using the NextSeq 500 High Output v2 Kit (Illumina) with a customized paired end, dual indexing (26/8/0/58-bp) format as recommended by 10× Genomics.

Cellranger (10XGenomics) was used to generate fastq files and count reads on each gene for each cell. Cellranger count output files were taken as input for R package seruat to perform single cell analysis. We first used illumina bcl2fastq v2.20.0.422 to demultiplex the raw sequencing data. Cellranger v2.1.1 (https://support.10xgenomics.com/single-cell-gene-expression/software/pipelines/latest/what-is-cell-ranger) was used to align raw fastq files to mm10 and perform barcode counting and UMI counting. The count matrix (column as cell; row as genes) from Cellranger count were taken as input for R package seruat v2.3 to perform single cell

analysis. Reads with the same UMI were combined and then annotated to ensemble genes (GRCm38/mm10). We filtered out the cells with <100 genes expressed and keep the cells with <15% mitochondria reads rate. The default setting of Seurat v2.3[62] was used to perform PCA. We used the first 20 principle components to perform cell cluster and t-SNE (resolution = 0.6). Markov Affinity-based Graph Imputation of Cells (MAGIC)[39] was used to perform the PCA and gene interaction analysis as guided by https://github.com/KrishnaswamyLab/MAGIC.

**WGBS library construction and data analysis.** Purified genomic DNA (with 5% of unmethylated lambda DNA spike-in, Promega) was sheared to till reaching a fragment size of 200–500 bp using Bioruptor UCD300 (Diagenode) according to manufacturer's instructions. Sheared DNA was ligated with methylated adaptors (NEBNext® Multiplex Oligos for Illumina®, NEB) by using a NEBNext® Ultra™ II DNA Library Prep Kit (NEB). Methylated adaptor-ligated DNA fragment was used for bisulfate conversion reaction with EZ DNA Methylation Kit (Zymo Research), then bisulfite converted DNA was amplified using KAPA HiFi HotStart Uracil + ReadyMix PCR Kit (Kapa Biosystems) with 8 cycles of PCR. Amplified DNA was purified by AMPureXP beads and examined by Agilent High Sensitivity DNA kit (Agilent Technologies) for quality check. Library concentration was determined by a Qubit 4 fluorometer (Thermo Fisher Scientific). Prepared libraries were sequenced using an Illumina NextSeq 500 instrument (150-cycle, paired-end).

Raw fastq files for WGBS (from E10.5-P0 stage) were downloaded from ENCODE. Raw fastq files were mapped to the hg19/mm10 genome assembly using bsmap-2.89 software with "-v 6 -n 1 -q 3 -r 1" parameters. The bisulfite conversion ratios were estimated using unmethylated lambda DNA. Mcall modual in MOABS[63] was used to call the mCG/CG ratios for each CpG site. Mcomp modual was used to call DMRs with parameter "–minNominalDif = 0.2–minDmcsInDmr 3–maxDistConsDmcs 500". The CpGs with coverage > = 5 was used for downstream analysis. The function prediction of DMRs was used for GREAT analysis[26]. UCSC genome browser tracks were generated by using the Mmint ucsc.py function.

**CMS-IP-seq library construction and data analysis.** CMS-IP-seq were performed as described previously with some modification[40,41]. Purified genomic DNA was sheared to yield 200–500 bp fragments using Bioruptor UCD300 (Diagenode) according to manufacturer's instructions. Bisulfite conversion was performed using the fragmented DNA with a EZ DNA Methylation-Lightning Kit (Zymo Research) to convert 5hmC to cytosine methyl sulfonate (CMS). CMS-containing DNA fragments were enriched using anti-CMS antibody and protein A/G dynabeads (Thermo Fisher Scientific). Enriched DNA fragments were then purified using the well-established phenol-chloroform-isoamyl alcohol extraction method. Purified DNA was then processed with a Pico Methyl-Seq Library Prep Kit (Zymo Research) to generate sequencing libraries. The quality of the DNA libraries was checked by an Agilent High Sensitivity DNA kit (Agilent Technologies), and then subjected to highthroughput sequencing on an Illumina NextSeq 500 instrument (75-cycle, single-end) (Supplementary Table 1).

Raw fastq data was mapped to the mm10/hg19 genome assembly using bsmap-2.89 with default parameters. After duplication removal, CMS peaks were called by using macs2 with default parameters. Bedtools merge was used to generate merged peaks for all samples. Reads numbers in each peak were counted if there is >1 bp overlap between reads location and peak region. The raw counts file with row as each peak, column as samples was used as input to DEGseq2 and differentially significantly CMS peaks (q value < = 0.05) between WT and Tet2/3-DKO were called. Volcano plots were plotted using R package ggplot2. The functions of decrease/increased CMS peaks between WT and DKO were predicted by using GREAT analysis[26]. Histone peaks regions and fold change over control bigwig files were downloaded from ENCODE (mm10 version). The average fold change of histone modifications signals over control on HypoDHMRs/HyperDHMRs and the average signal of CMS-IP-seq along metagenes were used with Mmint (https://github.com/lijiacd985/Mmint).

**ATAC-seq library construction and data analysis.** ATAC-seq library preparation was performed, as described previously[64]. Briefly, 50,000 cells were collected in ice-cold PBS. Nuclei were isolated in Cold Lysis Buffer (10 mM Tris–HCl, pH 7.4, 10 mM NaCl, 3 mM MgCl2, 0.1% IGEPAL). The transposition reaction was performed by using a Nextera DNA Library Preparation Kit (Illumina) with modified tagmentation condition (37 °C for 30 min). Tagmented DNA was purified by E.Z. N.A.® MicroElute Cycle Pure Kit (Omega BIO-TEK), then amplified with the KAPA real-time library amplification kit (Kapa Biosystems) followed by library purification using AmpuXP beads. The quality of purified DNA libraries was checked by Agilent High Sensitivity DNA kit (Agilent Technologies). The library was sequenced using an Illumina NextSeq 500 instrument (150 cycle, paired-end) (Supplementary Table 1).

Bowtie2 with '-very-sensitive' option was used to map the high-quality reads to mm10 version of human genome. The uniquely properly paired mapped reads were extracted for downstream analysis. MACS2 with the '-nomodel' and '-extsize 147′ was used to call ATAC peaks. Bedtools intersect (at least 1 bp overlap) was used to identify overlapped 5hmC peaks and ATAC peaks. We first cut the genome

to 10 kb equal size bins and use bigwigOverbed to calculate each bin's 5hmC and ATAC signals. R package geneplotter was used to plot the density scatterplots. The findMotifsGenome.pl in HOMER software was used for the motif enrichment with default setting. Randomly selected sequences from the genome with matched GC% content were used as background.

**Cleavage under targets and release using nuclease (Cut&Run)**. Cut&Run was performed according to published protocol[65]. Briefly, single cells were attached to the concanavalin A-coated magnetic beads (Bangs Laboratories) followed by the in situ binding of the antibody and pA-MN specifically to the target protein. Cleaved fragments were released after exposure to calcium. DNA was extracted from the supernatant containing released chromatin fragments. Libraries were prepared using the ThruPLEX DNA-seq Kit from Rubicon Genomics (R400406) according to the manufacture's instruction with slightly modification of changing extension time of library amplification steps to 20 s. Libraries were sequenced using the NextSeq 500 High Output v2 Kit (Illumina, San Diego, CA) with a customized paired end, dual indexing (40/8/0/40-bp) format (Supplementary Table 1).

We analyzed CUT&RUN data using the script on github (https://github.com/Henikoff/Cut-and-Run). Briefly, we first mapped paired end raw fastq files to mm10 use bowtie 2.2.5 with parameters "–local–very-sensitive-local–no-unal–no-mixed–no-discordant–phred33 -I 10 -X 700". Then we used picard to remove duplication reads; bamToBed was used to transform bam file to bed file. Next, we used spike_in_calibration.csh script to perform spike in normalization. BedGraphtobigwig was used to transform the bedGraph file to bigwig file, which was used to perform the visualization.

**HiChIP library construction and data analysis**. HiChIP library preparation was performed as described previously[49]. Briefly, 1 million crosslinked cardiac cells from mouse embryos or 2 million mouse embryonic stem cells were lysed with Hi-C Lysis Buffer (10 mM Tris–HCl pH 8.0, 10 mM NaCl, 0.2% NP-40 with 1× protease inhibitor cocktail), followed by digestion with 150 U of MboI restriction enzyme (NEB) for 2 h at 37 °C. Fill-in master mix containing biotin-dATP (Thermo Fisher Scientific) was added to digest nuclei to generate enzyme-digested overhang and mark the DNA ends with biotin. Subsequently, T4 DNA ligase (NEB) was added to the reaction and incubated for 4 h at room temperature to achieve proximity ligation. After that, nuclei were resuspended with Nuclear Lysis Buffer (50 mM Tris–HCl pH7.5, 10 mM EDTA, 1% SDS with 1× protease inhibitor cocktail) and transferred to Covaris millitube for fragmentation. Fragmented samples were precleared by adding protein A/G beads (Thermo Fischer Scientific) for 1 h at 4 °C, and 2 μg Smc1a antibody was added with an overnight incubation. On day 2, protein A beads (Thermo Fisher Scientific) were added to the reaction for 2 h to capture the beads. After bead capturing, the samples were washed three times each with low salt wash buffer (0.1% SDS, 1% Triton X-100, 2 mM EDTA, 20 mM Tris–HCl pH 7.5, 150 mM NaCl), high salt wash buffer (0.1% SDS, 1% Triton X-100, 2 mM EDTA, 20 mM Tris–HCl pH 7.5, 500 mM NaCl), and LiCl wash buffer (10 mM Tris–HCl pH 7.5, 250 mM LiCl, 1% NP-40, 1% sodium deoxycholate, 1 mM EDTA) at room temperature. After these steps, ChIP samples were resuspended in a DNA elution buffer (50 mM sodium bicarbonate pH8.0, 1% SDS) and incubated for 10 min at room temperature, followed by shaking for 3 min at 37 °C. DNA eluted from the beads were collected twice, followed by reverse crosslinking. Reverse crosslinked DNA were then purified by using an E.Z.N.A.® MicroElute Cycle Pure Kit (Omega BIO-TEK). Purified samples were used for biotin pull-down. Resuspended Streptavidin C-1 (Thermo Fisher Scientific) with 2× Biotin binding buffer (10 mM Tris–HCl pH7.5, 1 mM EDTA, 2 M NaCl) was added to the samples and proceed biotin capturing procedure by incubating 15 min at room temperature, followed by washes with Tween wash buffer (5 mM Tris–HCl pH7.5, 0.5 mM EDTA, 1 M NaCl, 0.05% Tween-20) and 1× TD buffer (10 mM Tris-HCl pH7.5, 5 mM magnesium chloride, 10% dimethylformamide), respectively. After wash, on-bead tagmentation by using Tn5 transposase (Illumina) was performed for 10 min at 55 °C with interval shaking, followed by several washes with 50 mM EDTA, Tween-20 wash buffer, and 10 mM Tris–HCl respectively. After wash, the reaction beads were resuspended in a PCR master mix (Q5® High-Fidelity 2× Master Mix, NEB, with Nextera Ad1.1 (Universal) and Ad2.X (barcoded) primers) for library amplification. Amplified on-bead DNA were eluted using a magnet and purified with an E.Z.N.A.® MicroElute Cycle Pure Kit (Omega BIO-TEK). The quality of libraries was checked by Agilent High Sensitivity DNA kit (Agilent Technologies). The library was sequenced on an Illumina NextSeq 500 instrument (150 cycle, paired-end) (Supplementary Table 1).

A total of 1 million cells from each condition were used to perform HiCHIP experiments. To improve the statistic power, we merged the two biological replicates to increase sequencing depth. HiC-Pro[66] was used to map the raw paired-end fastq files to mm10 genome assembly and identify the uniquely validated paired reads. build_raw_maps.sh and ice_norm.sh embeded in HiC-Pro pipeline were used to generate the raw contact map and normalized contact map. Fit-Hi-C[50] was used to identify the significant Paired End Tags (PETs) between any two bins (5 kb) with a $p$ value <0.05. We linked the PETs with the whole genic regions. The PETs linked to specific genes were counted and normalized, as shown in a previous study[49]. The RNA-seq, CMS-IP, ATAC-seq, ChIP-seq data and pairwise files containing PETs information were uploaded to WashU Epigenome Browser[67] (http://epigenomegateway.wustl.edu).

**Prediction of AB compartment**. The output file *_allValidPairs.hic from HiC-Pro pipeline was used as input file for juicer_tools.1.7.5_linux_x64_jcuda.0.8.jar eigenvector function[68]. The *_allValidPairs.hic file stores all the raw interaction paired reads between any two genomic bins from the same chromosome. The eigenvector for each chromosome with KR normalization at 50 kb resolution were calculated. For each chromosome, we manually checked the overlap between the compartment assignment and the accessible regions from our ATAC-seq data to decide if the compartment assignment need to be flipped.

**Chromatin immunoprecipitation quantitative PCR (ChIP-qPCR)**. ChIP was performed according to a previously described protocol with slight modifications[69]. Briefly, 20 million mESCs were fixed with 1% formaldehyde for 15 min at room temperature followed by quenching with 125 mM glycine for 5 min. Cells were washed using ice cold PBS twice and then resuspended in a sonication buffer (10 mM Tris pH 8.0, 0.25% SDS, 2 mM EDTA and protease inhibitor cocktail). The M220 Focused-ultrasonicator (Covaris) was used to sonicate the chromatin into 200–700 bp range. Pre-washed 25 μl protein G Dynabeads, 10 μg YY1 antibody and sheared chromatin were incubated overnight. The enrichment mixture was washed twice with each of the following buffers: RIPA-low salt (10 mM Tris HCl pH 8.0, 140 mM NaCl, 1 mM EDTA pH 8.0, 0.1% SDS, 0.1% Na-Deoxycholate, 1% Triton X-100 and protease inhibitor cocktail), RIPA-high salt (10 mM Tris HCl pH 8.0, 500 mM NaCl, 1 mM EDTA pH 8.0, 0.1% SDS, 0.1% Na-Deoxycholate, 1% Triton X-100 and protease inhibitor cocktail), RIPA-LiCl (10 mM Tris HCl pH 8.0, 250 mM LiCl, 1 mM EDTA pH 8.0, 0.1% SDS, 0.1% Na-Deoxycholate, 0.5% NP-40 and protease inhibitor cocktail) and TE buffer (10 mM Tris HCl pH8.0, 1 mM EDTA pH 8.0 and protease inhibitor cocktail). After). IP fragments were incubated at 55 °C for 1 h in elution buffer (10 mM Tris pH8.0, 5 mM EDTA pH 8.0, 300 mM NaCl, 0.4% SDS). The elution were reverse crosslinked at 65 °C overnight with 2 μl 20 mg/ml proteinase K. DNA fragments were purified using MicroElute Cycle-Pure Kit (OMEGA). qPCR was performed following the protocol described above.

**4C-seq library construction and data analysis**. 4C-seq was performed as previously described[47]. In total 10 million mESCs were crosslinked with 1% formaldehyde in PBS contains 10% FBS for 10 min. Glycine was added to a final concentration of 125 mM to quench the reaction. Cells were then washed twice using ice cold PBS followed by snap freezing with liquid nitrogen and stored at −80 °C. Ice cold Hi–C lysis buffer (10 mM Tris–HCl pH8.0, 10 mM NaCl, 0.2% Igepal and protease inhibitor cocktail) was used to isolate the nuclei. For the *Mpdu1* loci, DpnII (NEB R0543) was used for the primary digestion and BfaI (NEB R0568S) was used for the secondary digestion. PCR was performed using the Roche Expand Long Template polymerase (Roche 11759060001). Libraries were generated using the the NEB Next Ultra II DNA Library Prep Kit prior to sequencing (NEB #E7103). All the oligonucleotides are listed in Supplementary Table 2. Basic4Cseq R package was used to calculate normalized 4 C signals and plot 4C signals nearby the targeted regions.

**Reporting summary**. Further information on research design is available in the Nature Research Reporting Summary linked to this article.

## Data availability

The WGBS, CMS-IP, ATAC-seq, CUT&RUN, and RNA-seq (bulk and single-cell) data from this study have been submitted to the NCBI Gene Expression Omnibus (GEO; https://www.ncbi.nlm.nih.gov/geo/) under accession number GSE121671. All relevant data supporting the key findings of this study are available within the article and its Supplementary Information files or from the corresponding author upon reasonable request. The source data underlying Figs 1e, 2b–d, 4a, 6a, 7h and Supplementary Figs 3f and 8a are provided as a Source Data file. A reporting summary for this Article is available as a Supplementary Information file.

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

## Acknowledgements

We are grateful for Dr. Jianjun Shen and the MD Anderson Cancer Center next-generation sequencing core at Smithville (CPRIT RP120348 and RP170002), and the Epigenetic core in Institute of Biosciences and Technology at the Texas A&M University. This work was supported by grants from Cancer Prevention and Research Institute of Texas (RR140053 to Y.H., RP170660 to Y.Z., RP180131 to D.S.), the Innovation Award from American Heart Association (16IRG27250155 to Y.H.), the John S. Dunn Foundation Collaborative Research Award (to Y.H.), National Institute of Health grants (R01HL134780 to Y.H., R01HL146852 to Y.H., R01DE023177 to J.F.M., R01HL127717

to J.F.M., R01HL130804 to J.F.M., R01HL118761 to J.F.M., F31HL136065 to M.C.H., R01GM112003 to Y.Z., R01HL123953 to J.C.), the Welch Foundation (BE-1913-20190330 to Y.Z.), the American Cancer Society (RSG-18-043-01-LIB to YH; RSG-16-215-01-TBE to Y.Z.), Vivian L. Smith Foundation and MacDonald Research Fund Award (16RDM001 to J.F.M.), Transatlantic Network of Excellence Award LeDucq Foundation Transatlantic Networks of Excellence in Cardiovascular Research 14CVD01: "Defining genomic topology of atrial fibrillation." (J.F.M.), and by an allocation from the Texas A&M University start-up funds (Y.H. and D.S.).

## Author contributions

Y.H. and J.F.M. directed and oversaw the project. S.F. and Y.X. performed in vivo embryonic development studies. S.F., W.H., and T.L. maintained the genetically modified mice and performed histological analyses. S.F. and M.L. performed high-throughput sequencing library preparation. S.F. performed the CUT&RUN, dot-blot, ChIP-qPCR analysis. S.F. and T.H. performed Tet1CD rescue experiment. S.F. and L.G. performed 4 C analysis. J.L. and D.S. performed integrative data analysis. R.X. and P.Z. provided human embryo samples. M.H., W.M., F.W., J.C. and Y.Z. provided intellectual inputs. All the authors participated in discussion and data interpretation. Y.H., Y.Z. and J.F.M. wrote the manuscript.

## Additional information

**Competing interests:** The authors declare no competing interests.

