## [Peer Review File · Nature Communications]

Reviewers' comments:

Reviewer #1 (Remarks to the Author):

Fang and colleagues report that Tet2/3-mediated DNA demethylation plays an important role in early cardiac development. The authors employed genomic approaches and analyzed DNA methylation and hydroxymethylation dynamics, transcriptional reprogramming and high-order chromatin organization changes induced by Tet2/3 deletion. While the experiments and bioinformatics analysis look convincing, showing that TET2/3 depletion can lead to observations potentially through impaired DNA demethylation and chromatin accessibility at key genes during early cardiac development, the link between Tet2/3 depletion to the YY1 related long-range chromatin change is weak. This work provided data on Tet2/3-mediated chromatin change involved in early cardiac development, but additional experiments are required.

Main points:

1\ Figure 1 B, 5mC seems to undergo dynamic changes at different stages during embryonic cardiac development. Are most of the DMRs occurring at the same regions?

2\ Figure 1 C-D showed that 5mC undergoes dynamic changes especially at the cis-regulatory elements of key genes essential for heart function (i.e. Bmp10 and Tnnt2). H3K4me1 and H3K27ac ChIP-seq data could help to show these are cis-regulatory elements such as enhancers. The methylation level on the DMR upstream to Bmp10 promoter doesn't seem to change much.

3\ Figure S1A, It is better to provide western-blot results of Tet1/2/3. Gene expressions of Tet1/2 increase gradually during cardiac development, but not Tet3> does this mean Tet1/2 play more important roles than Tet3?

4\ Figure 3A, what's the cutoff wfor the differentially expressed genes? It is not common to identify so many DEGs regulated by Tet2/3 depletion, especially at E12.5 embryos. How many of these genes are related to 5mC and 5hmC changes?

5\ Figure 5J, It seems that YY1 binding is disrupted on 5hmC unmodified regions, and the YY1 binding density of almost all the YY1 peaks is lower in TKO cells than WT cells. How are the 5mC levels on these YY1 peaks? Is YY1 binding generally down-regulated in TKO mESCs?

6\ Figure 6G, HiChIP experiment needs to be in replicates. Figure S8E-F, TET protein rescue in TKO mESCs is needed to show it can rescue the reduction of PETs at YY1 enriched regions in Tet-TKO mESCs compared to the WT ESCs.

Reviewer #2 (Remarks to the Author):

In this manuscript, Fang and Li et al investigate the role of Ten-Eleven Translocation protein 2 and 3 in cardiac development and regulation of the epigenome. Through multiple datasets, they argue that Ying-Yang 1 (YY1) binding is impaired upon deletion of Tet2 and 3 in cardiac cells, which results in loss of proper genome organization.

Overall, the manuscript is well-written, and the work is well-presented. The number datasets in this manuscript is particularly impressive. However, at times the datasets are not well-explored/described in the manuscript (one example being the single cell RNA-expression datasets) and felt like the authors focused on the particular points from the datasets which were consistent with their model. The YY1 data is interesting, though similar findings have been published recently by two other groups (<https://www.ncbi.nlm.nih.gov/pubmed/28536180> and [https://www.cell.com/cell/pdf/S0092-8674\(17\)31317-X.pdf](https://www.cell.com/cell/pdf/S0092-8674(17)31317-X.pdf)). The Tet-YY1 connection is interesting, though two important points are somewhat glossed over: 1. How is the Tet-YY1 connection made (is YY1 recruited to DNA-methylated regions/by TET, etc?), and 2. How exactly the phenotype of the Tet KO mice relates to some of the epigenetic differences observed is unclear. Clearly, Tet enzymes control the expression of many important genes in the heart, but it is not clear if the ATAC-seq differences and/or differences in HiChIP are causative of the gene expression changes. Some additional data regarding controls, results of ESC experiments, and embryo analysis is required to boost the interpretations.

Additional concerns

It would be helpful to report basic statistics regarding the hypomethylated regions in 1B and 4C (median/mean size, %genome coverage), so the data can be compared to similar other studies.

Regarding the GREAT/GO analysis (Fig 1C/D, 4D/E, 6D) – are these the only categories that appear or others are represented as well?

The genotyping table in Figure 2 needs to be more fully developed. For example, it's not clear what proportion of animals were expected to be "Het" (what genotype is being considered as het?). How many litters were analyzed in the late stages? Better annotating the table would be helpful to the reader so it's not necessary to refer to the figure legend. Related to this point, The DORV in S3D is not well-shown. The RV OFT is not well observed in the images provided. Also, this is a bit surprising, as this phenotype is usually related to abnormalities in the neural crest. Were any observed (migration, survival of neural crest in the OFT cushions?)

The rationale for the single cell studies is not well-presented. Were the authors expecting greater heterogeneity of expression upon loss of Tet2/3? For example, the Rao group recently reported that TET proteins balance neural vs cardiac differentiation (<https://www.ncbi.nlm.nih.gov/pmc/articles/PMC5187696/>). Were the authors expecting persistence of genes related to neural biology in the mutants which might have been obscured in the bulk RNA-sequencing datasets? In addition, it's hard to judge the quality of the datasets without a figure of the clustering and the transcriptional signatures used to define the cell types explored in Figure 3D. Also, information about how many cells were sequenced from how many replicates, depth of sequencing, transcripts per cell would be useful. The reduction of cardiac myocytes is not entirely surprising based on the histology – so why the single cell approach? Was there a signal for a reduction in Tbx20/Ttn expression in the bulk RNA-seq?

Is the reduction in 5hmc observed in Figure 4A/B due to a general reduction in 5hmc in all cells in the mutant or isolated reduction of 5hmc in some cells?

The studies related to Tet-YY1 are interesting. The ESCs presented were generated using CRISPR-based approaches. The data demonstrating they are null should be presented. Also, a potential caveat with this approach, as opposed to an inducible deletion approach, is that over passages, the KO and control cell lines acquire differences – and thus making it appear there is a difference in YY1. Hence, an inducible approach to either delete (or not delete) the enzymes in the same "starting cell" would be more definitive.

Is there a reduction in chromatin-bound YY1 in the Tet 2/3 DKO mice? This is an especially important point as the ESC experiments were performed using Tet1/2/3 KO cells.

The A-to-B switch upon loss of Tet2/3 in cardiac tissues. Since the B compartment is enriched for Lamina-bound chromatin, was there a change in spatial positioning of these genes relative to the lamina? What about B-to-A switching? This is relevant to YY1 as well since it was shown that YY1 might instruct lamina-associated chromatin binding (<http://jcb.rupress.org/content/208/1/33>).

Minor points

Statistics on 1E would be helpful.

The reduction in CMS-IP and ATAC-seq signal is impressive in 5E. An example of where those signals are not as reduced in the mutant condition would be helpful. Same for 6G

The tracks in 5J are nice, but it would be also nice to show where the peak calls are in each of the conditions. Related to this, are the differences in S7F statistically significant? It's not noted in the figure or the legend.

Typo in Figure 7 (Chromatinc accessibility)

Reviewer #3 (Remarks to the Author):

This study by Fang et al first examines the dynamics of DNA methylation and demethylation in early cardiac development. The authors then leverage this information to examine how TET enzymes that are involved in demethylation and regulate cardiac development, observing significant congenital heart disease when Tet2/3 are disrupted in early embryos. Finally, the authors utilize integrated epigenomic and RNA-seq datasets to infer a critical role for the transcription factor YY1 along with Tet-dependent changes in methylation as mediators of normal cardiac differentiation. The study is well-conceived and significant, as this is an area of fundamental biologic importance in developmental biology and cardiovascular disease. Overall, I believe the conclusions are based on solid experimental evidence.

Major Questions:

1) The authors determine that Tet inactivation results in disruption of YY1 DNA binding based on GREAT computational analysis of ATAC-seq data from mouse DKO hearts. Can the authors demonstrate that YY1 DNA binding is altered during cardiac development at DMRs in the Tet2/3 DKO mouse model using CUT and RUN?

2) The authors use mESC to study YY1 DNA binding in a triple Tet KO model. However, it does not appear that the ESCs were differentiated toward a cardiomyocyte lineage. Can the authors demonstrate differential binding of YY1 in mESC during differentiation into cardiomyocytes using DKO? It would be ideal to knockout Tet2/3 at the cardiac progenitor stage and then examine how YY1 DNA binding changes. Alternatively, can the authors demonstrate that these regions function as enhancers, using an exemplary locus.

3) I don't fully understand how motif enrichment was determined in Figure S6F. Can the authors clarify whether this enrichment is based on motif analysis at ATAC-seq peaks compared to random nucleotide background?

4) The authors use Nkx2.5 as a model for cardiomyocyte-specific deletion of the Tet enzymes. However, it is possible that this Cre model will drive loss of Tet2/3 expression in non-cardiomyocyte populations (e.g. fibroblasts or endothelial cells/vascular smooth muscle cells) that could affect cardiac development. The authors need to address this aspect of the model in their discussion to acknowledge the possibility for non-cell autonomous effects of Tet inactivation in vivo.

Response to Comments:

We thank reviewers for their constructive suggestions, which we feel are very important to further improve our manuscript. To fully address the prior concerns and comments raised by reviewers, we have performed the recommended additional experiments and analyses. We have incorporated new data (Fig. 1, 2, 5, 6, 7, and Fig. S1, S3, S5, S6, S7, S8 and S9; See the attached **Table** below) and the major changes made in the texts were shown in blue.

Reviewer #1 (Remarks to the Author):

Fang and colleagues report that Tet2/3-mediated DNA demethylation plays an important role in early cardiac development. The authors employed genomic approaches and analyzed DNA methylation and hydroxymethylation dynamics, transcriptional reprogramming and high-order chromatin organization changes induced by Tet2/3 deletion. While the experiments and bioinformatics analysis look convincing, showing that TET2/3 depletion can lead to observations potentially through impaired DNA demethylation and chromatin accessibility at key genes during early cardiac development, the link between Tet2/3 depletion to the YY1 related long-range chromatin change is weak. This work provided data on Tet2/3-mediated chromatin change involved in early cardiac development, but additional experiments are required.

Main points:

1. Figure 1 B, 5mC seems to undergo dynamic changes at different stages during embryonic cardiac development. Are most of the DMRs occurring at the same regions?

To address this, we performed further analyses on DMRs and confirmed that the majority of DMRs (99%) were stage specific and occurred at different genomic regions for each development stage. Only 1% of common DMRs were shared among these developmental stages. A new panel has been added (New Figure S1B), with the related description/discussion added in the second paragraph of the section entitled “*5mC and 5hmC undergo dynamic changes during embryonic heart development in both human and mice*”.

“Notably, very few DMRs (1%) were commonly shared among the analyzed developmental stages (Figure S1B), and the majority of DMRs (~99%) were identified at different genomic regions for each developmental stage. This finding suggests that the observed focal DNA methylation changes are stage-specific, rather than occurring at the same genomic regions, during embryonic heart development.”

2. Figure 1 C-D showed that 5mC undergoes dynamic changes especially at the cis-regulatory elements of key genes essential for heart function (i.e. Bmp10 and Tnnt2). H3K4me1 and H3K27ac ChIP-seq data could help to show these are cis-regulatory elements such as enhancers. The methylation level on the DMR upstream to Bmp10 promoter doesn't seem to change much.

We have compared the DNA methylation level, H3K4me1 and H3K27ac enrichment in selected Bmp10 and Tnnt2 regions. We observed strong enrichment of H3K4me1 and H3K27ac peaks at these two selected regions. Due to the space limitation, we only showed the H3K27ac enrichment at these regions in Figure 1D.

In the original figure, because of the over-crowded, small size of genomic region in the genome browser views, it gives the false impression that “*The methylation level on the DMR upstream to Bmp10 promoter doesn’t seem to change much*”. To more accurately compare the methylation level on the DMR upstream to the Bmp10 promoter, we calculated the DNA methylation level at the highlighted Bmp10 enhancer (as in Figure 1D; top panel-red traces) and replotted the data in the appended Figure 1 (Figure A1, for reviewers only): the DNA methylation at this region indeed showed a gradual decrease during embryonic heart development (from 0.43 at E10.5 to 0.19 at P0).

3. Figure S1A, It is better to provide western-blot results of Tet1/2/3. Gene expressions of Tet1/2 increase gradually during cardiac development, but not Tet3> does this mean Tet1/2 play more important roles than Tet3?

We thank the reviewer for pointing out this. Based on our experience using embryonic stem cells, it requires a significant number of cells (>10 million) to perform WB analysis of Tet1/2. Given the limited cell numbers obtained from heart tissues at different developmental stages, we did not perform WB to detect Tet1/2 protein levels. For Tet3, the lack of WB-grade antibody prevented us detecting endogenous Tet3 expression. However, we presented alternative evidence to show Tet1/2/3 expression at the mRNA levels (Figure S1C), and also performed IHC staining to detect Tet1/2 expression at tissue levels (Figure S1E). We hope that the reviewer will understand the technical difficulties for this experiment.

Although the expression of *Tet1* and *Tet2*, but not *Tet3*, undergoes dynamic changes in cardiac tissues during embryonic development, the deletion of *Tet1* or *Tet2* alone in mice did not cause overt cardiac developmental defects (PMID: 21816367, PMID: 21873190). Deletion of *Tet3* resulted in embryonic lethality in mice (PMID: 21892189), suggesting that *Tet3* indeed plays an indispensable role in regulating embryonic development. Hence, more studies are required before we can claim that “*Tet1/2 play more important roles than Tet3*”.

4. Figure 3A, what’s the cutoff for the differentially expressed genes? It is not common to identify so many DEGs regulated by Tet2/3 depletion, especially at E12.5 embryos. How many of these genes are related to 5mC and 5hmC changes?

We used cuffdiff2 with the default setting (FDR<=0.05) to identify differentially expressed genes (DEGs) in Figure 3A. If we used fold change >= 2 & FDR<=0.05 (a more stringent cutoff) as the threshold, the DEG numbers were: 497 (174 up- and 323 down-regulated in DKO) and 291 (156 up- and 135 down-regulated in DKO) at E12.5 and E15.5 stages, respectively.

To correlate DEGs with changes in 5mC or 5hmC, we re-analyzed 5hmC and 5mC levels at genomic regions 2kb up- and down-stream of 497 DEGs identified at E12.5. It turned out that 43.1% of the downregulated DEGs showed 5hmC decrease, and 39.8% of the downregulated DEGs showed gain of 5mC (NEW Figure S5K). This is consistent with several previous reports demonstrating that not all the 5mC/5hmC changes are associated with gene expression (PMID: 24474761, PMID: 21460036). We have added the related data in the NEW Figure S5K, along with the related description in the third paragraph of the section “*Tet2/3 depletion causes a global decrease of 5hmC in embryonic hearts*”.

5. Figure 5J, It seems that YY1 binding is disrupted on 5hmC unmodified regions, and the YY1 binding density of almost all the YY1 peaks is lower in TKO cells than WT cells. How are the 5mC levels on these YY1 peaks? Is YY1 binding generally down-regulated in TKO mESCs?

As shown in the NEW Figure 6F and NEW Figure S8I, we observed that the majority of YY1 binding regions displayed increased DNA methylation in TKO mESCs or DKO CMs compared with their parental controls. As suggested by the reviewer, we compared YY1 binding in control and TKO mESC, there are 73% (10,984 out of 15,055) of YY1-enriched regions showed decreased binding in Tet-TKO mESC (NEW Figure S8D). A similar trend was noted when comparing heart tissues from control and Tet-DKO mice (NEW Figure S8G). We have added the related data and description in NEW Figure 6F, S8 and the third paragraph in the section “5hmC loss compromises YY1 binding to the genome by modulating chromatin accessibility in embryonic heart tissues”.

6. Figure 6G, HiChIP experiment needs to be in replicates.

As shown in Figure S9A, the HiChIP experiment was performed with biological duplicates.

7. Figure S8E-F, TET protein rescue in TKO mESCs is needed to show it can rescue the reduction of PETs at YY1 enriched regions in Tet-TKO mESCs compared to the WT ESCs.

We performed the rescue experiment in Tet TKO mESCs by re-expressing Tet1CD (the catalytic domain for Tet1). We confirmed Tet1CD expression, as well as 5hmC production in the reconstituted cells (NEW Figure S9H-I). We then moved on to use Tet-TKO mESCs, with and without Tet1CD expression, to examine the YY1 binding and chromatin looping at the *Mpdu1* locus using YY1 ChIP-qPCR and 4C-seq, respectively. Indeed, Tet protein expression in Tet-TKO mESCs could effectively restore YY1 binding and chromatin looping at the *Mpdu1* locus (NEW Figure 7H-I). Overall, the new data strongly support the conclusion that Tet / 5hmC contributes to the YY1 mediated long-range chromatin looping.

Reviewer #2 (Remarks to the Author):

Overall, the manuscript is well-written, and the work is well-presented. The number datasets in this manuscript is particularly impressive. However, at times the datasets are not well-explored/described in the manuscript (one example being the single cell RNA-expression datasets) and felt like the authors focused on the particular points from the datasets which were consistent with their model. The YY1 data is interesting, though similar findings have been published recently by two other groups (<https://www.ncbi.nlm.nih.gov/pubmed/28536180> and [https://www.cell.com/cell/pdf/S0092-8674\(17\)31317-X.pdf](https://www.cell.com/cell/pdf/S0092-8674(17)31317-X.pdf)). The Tet-YY1 connection is interesting, though two important points are somewhat glossed over: 1. How is the Tet-YY1 connection made (is YY1 recruited to DNA-methylated regions/by TET, etc?), and 2. How exactly the phenotype of the Tet KO mice relates to some of the epigenetic differences observed is unclear. Clearly, Tet enzymes control the expression of many important genes in the heart, but it is not clear if the ATAC-seq differences and/or differences in HiChIP are causative of the gene expression changes. Some additional data regarding controls, results of

ESC experiments, and embryo analysis is required to boost the interpretations.

The connection between Tet and YY1 was initially made on the basis of our unbiased integrative epigenome analysis. We performed motif analysis in genomic regions showing decreased 5hmC and chromatin accessibility in Tet2/3-DKO embryonic heart tissues (Figure 5F). YY1 binding regions emerged as the top candidate, and naturally became our focus for further characterization. Next, we performed YY1 chromatin binding and CUT&RUN experiments in control and Tet-TKO mESCs, and confirmed that Tet deletion indeed led to a reduction in YY1 binding (Figure 6A, D, NEW Figure S8D), thus establishing a functional link between Tet and YY1 binding.

To explore the possibility that YY1 might interact with Tet proteins for genomic binding, we further performed Co-IP experiments in mESC, but failed to detect any interaction (Figure A3 for reviewers only). Therefore, it is less likely that “YY1 (is) recruited to DNA-methylated regions/by TET”. At least, this is not the case for TET1 (point 1 by the reviewer).

To address Point 2 (link with epigenetic differences), we compared YY1 enrichment with ATAC-seq (chromatin accessibility) and the DNA methylation levels between control and Tet2/3-DKO E12.5 heart tissues (Figure A4 for reviewers only). 47% (1,764 out of 3,776) of chromatin accessible regions displayed YY1 enrichment; while only 3% (113 out of 3,600) of DMRs overlapped with YY1 enriched region. Furthermore, the majority of regions (84%, 2,362 out of 2,816) with reduced chromatin accessibility displayed decreased YY1 binding in the Tet2/3-DKO group (NEW Figure 6H). Based on these data, our current model is that Tet deletion leads to a reduction in the chromatin accessibility to prevent the binding of YY1 to its genomic targets.

Overall, we have performed the essential experiments recommended by the reviewer with proper controls and detailed re-analysis of the data. Admittedly, based on the current data, it is difficult to pinpoint the exact progenitor cells or developmental pathways that are affected by Tet loss-of-function during heart development. We are currently working on this by using more extensive single-cell analysis and lineage-tracing techniques. We hope that the reviewer will agree with us that such endeavor by itself will be beyond the scope of this study.

Additional concerns

3. It would be helpful to report basic statistics regarding the hypomethylated regions in 1B and 4C (median/mean size, %genome coverage), so the data can be compared to similar other studies.

We have followed the reviewer's advice and analyzed the hypomethylated regions in murine cardiac tissues during embryonic development. We have added the related results in the NEW Figure S1A. The red line indicated the mean size of hypomethylated DMRs and the genome coverage is approximately 0.01%. The mean size of hypoDHMRs identified from Figure 4C is 593 bp (0.18% genome coverage). We incorporated these numbers in the figure legend of Figure 4C.

4. Regarding the GREAT/GO analysis (Fig 1C/D, 4D/E, 6D) – are these the only categories that appear or others are represented as well?

The GREAT analysis revealed that the most represented components were significantly related to cardiac development and functions. Due to the space limitation, we selected the top categories tightly relevant to heart function. We have shown the entire list of GREAT/GSEA analysis in Figure A5 (for reviewers only). Meanwhile, we revised the figure legend to indicate that these were representative categories.

5. The genotyping table in Figure 2 needs to be more fully developed. For example, it's not clear what proportion of animals were expected to be "Het" (what genotype is being considered as het?). How many litters were analyzed in the late stages? Better annotating the table would be helpful to the reader so it's not necessary to refer to the figure legend. Related to this point, The DORV in S3D is not well-shown. The RV OFT is not well observed in the images provided. Also, this is a bit surprising, as this phenotype is usually related to abnormalities in the neural crest. Were any observed (migration, survival of neural crest in the OFT cushions?)

We have updated Figure 2A with detailed breeding information. We have also updated Figure S3D with the DORV phenotype. Our data indicate that the pulmonary trunk and aorta are side by side in a fashion typical of DORV. In addition, Dr. Martin's lab, as well as other labs have previously shown that DORV can occur from genetic defects in second heart field (SHF) development (PMID: 16836994, 21145505). Importantly, the *Nkx2.5cre* lineage includes the SHF (PMID: 17519332, 12141429). Although we do not detect strong evidence of neural crest defects, such as lack of separation of pulmonary trunk and aorta, we cannot rule out that there are non-autonomous mechanisms that impact cardiac neural crest development.

6. The rationale for the single cell studies is not well-presented. Were the authors expecting greater heterogeneity of expression upon loss of *Tet2/3*? For example, the Rao group recently reported that TET proteins balance neural vs cardiac differentiation (<https://www.ncbi.nlm.nih.gov/pmc/articles/PMC5187696/>). Were the authors expecting persistence of genes related to neural biology in the mutants which might have been obscured in the bulk RNA-sequencing datasets? In addition, it's hard to judge the quality of the datasets without a figure of the clustering and the transcriptional signatures used to define the cell types explored in Figure 3D. Also, information about how many cells were sequenced from how many replicates, depth of sequencing, transcripts per cell would be useful. The reduction of cardiac myocytes is not entirely surprising based on the histology – so why the single cell approach? Was there a signal for a reduction in *Tbx20*/*Ttn* expression in the bulk RNA-seq?

We used single-cell analysis because of the heterogeneity of cell composition in embryonic heart tissues. Using single-cell analysis, we hope to identify additional genes or cell subtypes that are affected by *Tet* deletion during early heart development. In our manuscript, we provided an example by using the MAGIC algorithm (Figure 3F-G). In addition to confirm our histological analysis results, we observed distinct expression pattern of *Tbx20* and *Hey2* in the control (*Tbx20^{high}Hey2^{high}*) and *Tet2/3*-DKO *Ttn⁺* CMs (*Tbx20^{low}Hey2^{low}*), which might reflect the differential maturation status of CMs. This information cannot be obtained from a bulk RNA-seq analysis.

As suggested by the reviewer, we have modified Table S1 to specify the single-cell RNA-seq data. We sequenced around 2,500-4,500 cells. Each cell was sequenced with 30,000-70,000 reads, which covered 1,500 to 2,500 genes. Overall, these parameters met the recommended coverage provided by the 10X Genomics single-cell RNA-seq guideline. The transcriptional

signatures used to define the cell types explored in Figure 3D were shown in Figure S4G and Table S3.

Due to the heterogeneity of analyzed cells, we might miss genes that are essential for cardiac development within the myocardium population if solely relying on bulk RNA-seq data. Indeed, as mentioned by the reviewer, we did not observe a significant differential expression of *Tbx20* in bulk RNA-seq analysis between the control and DKO group (Figure A2 for reviewer only); however, from the single-cell analysis, we observed a strong reduction of *Tbx20* in selected CMs (Figure 3E). With regard to the *Ttn* gene, we observed reduction of *Ttn* expression in both bulk and single-cell RNA-seq analysis (Figure A2 for reviewer only, Figure 3E). Together, the single-cell RNA-seq approach is able to provide additional insights into how Tet proteins regulate transcription within specific cardiac cell types.

We agree that the observations made by the Rao group are quite interesting. In our studies, we deleted Tet genes at stages subsequent to the cell fate transitions that were disrupted in the paper by the Rao group; and therefore; we did not see similar phenotypes. Please note that the Rao's group used different Cre drivers (*Zp3-Cre* and *Stra8-Cre*) to generate Tet deficient mice, which led to abrogation of Tet in selected tissues at very early stages (oocytes and sperm). We focused on elucidating how Tet proteins regulate the cellular composition and transcription of cardiac tissues, rather than the balance between neuronal and cardiac cell fate determination.

7. Is the reduction in 5hmC observed in Figure 4A/B due to a general reduction in 5hmC in all cells in the mutant or isolated reduction of 5hmC in some cells?

Our analysis performed in Figure 4A/B used the whole heart tissue. From our analysis, we observed a strong 5hmC reduction in heart tissue. However, there are still residual detectable 5hmC based on immunofluorescent and dotblot analyses. This might be due to the expression of Tet1 and/or the residual 5hmC in other non-CM tissues. We have added the related justification in the first paragraph in the section entitled "*Tet2/3 depletion causes a global decrease of 5hmC in embryonic hearts*".

8. The studies related to Tet-YY1 are interesting. The ESCs presented were generated using CRISPR-based approaches. The data demonstrating they are null should be presented. Also, a potential caveat with this approach, as opposed to an inducible deletion approach, is that over passages, the KO and control cell lines acquire differences – and thus making it appear there is a difference in YY1. Hence, an inducible approach to either delete (or not delete) the enzymes in the same "starting cell" would be more definitive.

We thank the reviewer for the great suggestions. The *Tet*-null background was confirmed by loci-specific sequencing (at the nucleic acid level), immunoblot (at the protein level) and dotblot for 5hmC (at the function level) analyses (Figure S7A-B, NEW Figure S7C). We agree that cell passage might affect YY1 distribution. The control cell we used was mESC expressing Cas9 without sgRNA targeted to the *Tet1/2/3* loci. Therefore, the control and TKO mESC share a similar passage history and treatment. While an inducible triple gene deletion strategy is an elegant design, it is a complex experiment to set up within the revision period. Moreover, our current experiments are well controlled. We have updated the methods section to clarify these points.

9. Is there a reduction in chromatin-bound YY1 in the Tet 2/3 DKO mice? This is an especially important point as the ESC experiments were performed using Tet1/2/3 KO cells.

To address the reviewer's concerns, we performed YY1 Cut & Run experiment in both the control and Tet2/3 DKO E12.5 heart tissues. As shown in the NEW Figure 6G and Figure S8G-I, consistent with the mESC data, we observed a global reduction of YY1 binding in Tet2/3-DKO heart tissues. We also observed decreased YY1 binding at genomic regions displaying reduced chromatin accessibility in Tet2/3-DKO heart tissues (NEW Figure 6H). The related data and description have been added in NEW Figure 6G-H, S8G-I and the third paragraph of a section entitled "*5hmC loss compromises YY1 binding to the genome by modulating chromatin accessibility in embryonic heart tissues*".

10. The A-to-B switch upon loss of Tet2/3 in cardiac tissues. Since the B compartment is enriched for Lamina-bound chromatin, was there a change in spatial positioning of these genes relative to the lamina? What about B-to-A switching? This is relevant to YY1 as well since it was shown that YY1 might instruct lamina-associated chromatin binding (<http://jcb.rupress.org/content/208/1/33>).

We thank the reviewer for pointing out the connection between YY1 and lamina associated chromatin regions. Since the current study is primarily focused on studying how TET/5hmC regulates YY1 binding, our logic is to first analyze the association between 5hmC and lamina-associated regions. Our assumption is that if we can identify 5hmC-enriched lamina-associated regions, we will move on to explore the regulation of YY1 within these regions. We analyzed the 5hmC peaks within lamina-enriched regions using the published lamin B ChIP-seq data (GSE97878) in E12.5 heart tissue. Unfortunately, there is no overlap between lamin B (n = 120,000) enriched regions and reduced 5hmC peaks (Tet2/3-DKO vs Control) (n = 8,846) identified from Tet2/3-DKO E12.5 heart tissue. This is largely consistent with our observation that 5hmC is enriched in compartment A, but not in compartment B. Based on this finding, it is less likely that YY1 is involved in instructing lamina-associated regions in our experiments. However, since only 16% of YY1 enriched regions overlapped with 5hmC peaks, we still cannot rule out the possibility that other YY1-bound regions (not overlapped with 5hmC enriched regions) might be involved in regulating A-to-B switch; but this topic is beyond the scope of this study.

Minor points

11. Statistics on 1E would be helpful.

Corrected as advised.

12. The reduction in CMS-IP and ATAC-seq signal is impressive in 5E. An example of where those signals are not as reduced in the mutant condition would be helpful. Same for 6G

Added as recommended. Typical examples of genome browser views have been added to the NEW Figure S6G (for the original Figure 5E, NEW Figure 6E) and S9E (for the original Figure 6G, NEW Figure 7G). Examples were given where 5hmC, ATAC-seq and promoter-enhancer looping signals were not reduced under the mutant condition.

13. The tracks in 5J are nice, but it would be also nice to show where the peak calls are in each of the conditions. Related to this, are the differences in S7F statistically significant? It's not noted in the figure or the legend.

By following the reviewer's advice, we have corrected all these concerns. We have updated NEW Figure 6E (original Figure 5J) with black bar indicating the peak calls. We have also added the p value ($p < 2.2 \times 10^{-16}$) for the updated NEW Figure S8F (original Figure S7F).

14. Typo in Figure 7 (Chromatinc accessibility).

Corrected. Due to the space constraint, we have simplified this figure as NEW Figure 7J.

Reviewer #3 (Remarks to the Author):

This study by Fang et al first examines the dynamics of DNA methylation and demethylation in early cardiac development. The authors then leverage this information to examine how TET enzymes that are involved in demethylation and regulate cardiac development, observing significant congenital heart disease when Tet2/3 are disrupted in early embryos. Finally, the authors utilize integrated epigenomic and RNA-seq datasets to infer a critical role for the transcription factor YY1 along with Tet-dependent changes in methylation as mediators of normal cardiac differentiation. The study is well-conceived and significant, as this is an area of fundamental biologic importance in developmental biology and cardiovascular disease. Overall, I believe the conclusions are based on solid experimental evidence.

Major Questions:

1) The authors determine that Tet inactivation results in disruption of YY1 DNA binding based on GREAT computational analysis of ATAC-seq data from mouse DKO hearts. Can the authors demonstrate that YY1 DNA binding is altered during cardiac development at DMRs in the Tet2/3 DKO mouse model using CUT and RUN?

We have followed the reviewer's suggestion to perform CUT&RUN analyses on both the control and Tet2/3 DKO heart tissues. We examined the changes in chromatin accessibility or DNA methylation within the YY1 enriched regions in both control and Tet2/3 DKO heart tissues. As shown in Figure A4 (for reviewer only), 47% (1,764 out of 3,776) of genomic regions with decreased chromatin accessible regions overlapped with YY1 enriched genomic regions; while only 3% (113 out of 3,600) of differential methylated regions (DMRs) overlapped with YY1-enriched regions. These data suggest that DNA methylation might not affect YY1 binding, and that Tet/5hmC-associated chromatin accessibility alterations might contribute more to YY1 binding.

2) The authors use mESC to study YY1 DNA binding in a triple Tet KO model. However, it does not appear that the ESCs were differentiated toward a cardiomyocyte lineage. Can the authors

demonstrate differential binding of YY1 in mESC during differentiation into cardiomyocytes using DKOs? It would be ideal to knockout Tet2/3 at the cardiac progenitor stage and then examine how YY1 DNA binding changes. Alternatively, can the authors demonstrate that these regions function as enhancers, using an exemplary locus.

We thank the reviewer for the helpful suggestions. As shown in Figure 6E, we used the ENCODE datasets to examine histone enrichment at the *Gja* and *Hey2* loci, we observed that within many YY1-enriched regions, we observed enrichment of enhancer markers, H3K27ac and H3K4me1. Due to technical difficulties and given the time limit, we were unable to achieve a high differentiation efficiency from mESC toward cardiomyocytes. However, we performed rescue experiments to demonstrate that re-expressing Tet1 in Tet TKO mESC could restore YY1 binding, as well as long-range chromatin looping (NEW Figure 7H-I, NEW Figure S9H-I).

3) I don't fully understand how motif enrichment was determined in Figure S6F. Can the authors clarify whether this enrichment is based on motif analysis at ATAC-seq peaks compared to random nucleotide background?

We used the "findMotifsGenome.pl in HOMER software" with the default setting to do the motif enrichment. By default, randomly selected sequences from the genome with matched GC% content are used as background. We have added this description in the method section entitled "ATAC-seq library construction and data analysis".

4) The authors use Nkx2.5 as a model for cardiomyocyte-specific deletion of the Tet enzymes. However, it is possible that this Cre model will drive loss of Tet2/3 expression in non-cardiomyocyte populations (e.g. fibroblasts or endothelial cells/vascular smooth muscle cells) that could affect cardiac development. The authors need to address this aspect of the model in their discussion to acknowledge the possibility for non-cell autonomous effects of Tet inactivation in vivo.

We agree that non-cell autonomous effects might influence the cardiac development and contribute to the phenotypes we observed in Tet-DKO mouse model. We have added related discussions on this point in the revised manuscript (page 13).

See the Appendix Table below for changes made in the figures (both main figures and SI figures).

Figures	Revised figures	In Response to Reviewer(s)
Figure. 1	Modified Figure 1D and 1E	R1 (comment 2) and R2 (comment 11).
Figure. 2	Modified Figure 2A	R2 (comment 5)
Figure. 5	Move Figure S6F to Figure 5F	R2 (comment 1)
NEW Figure 6 (Split Figure 5 into Figure 5 and Figure 6)	Modified New Figure 6E (Original Figure 5J)	R3 (comment 2)
NEW Figure 6	NEW Figure 6F	R1 (comment 5)
NEW Figure 6	NEW Figure 6G	R2 (comment 9) R3 (comment 1)
NEW Figure 6	NEW Figure 6H	R2 (comment 1, 9) R3 (comment 1)
Figure 7	Modified Figure 7J (Original Figure 7)	R2 (comment 14)
Figure 7	NEW Figure 7H-I	R1 (comment 7) R3 (comment 2)
Supplementary Fig. S1	Added new Figure S1A	R2 (comments 3)
Supplementary Fig. S1	Added new Figure S1B	R1 (comments 1)
Supplementary Fig. S3	Modified Figure S3D	R2 (comments 5)
Supplementary Fig. S5	Added New Figure S5K	R1 (comments 4)
Supplementary Fig. S6	Added New Figure S6G	R2 (comments 12)
Supplementary Fig. S7	Added New Figure S7C	R2 (comment 8)
Supplementary Fig. S8 (Split Figure S7 into S7 and S8)	Modified Figure S8D (Original Figure S7F)	R2 (comment 13)

Supplementary Fig. S8	Added New Figure S8E	R1 (comment 5) R2 (comment 1, 9)
Supplementary Fig. S8	Added New Figure S8G	R2 (comment 9) R3 (comment 1)
Supplementary Fig. S8	Added New Figure S8H-I	R1 (comment 5)
Supplementary Fig. S9	Added New Figure S9E	R2 (comment 12)
Supplementary Fig. S9	Added New Figure S9H-I	R1 (comment 7) R3 (comment 2)
Table S1	Modified Table S1 with single cell analysis	R2 (comment 6)
Table S2	Modified Table S2 with additional primers for 4C-seq	R1 (comment 7) R3 (comment 2)
Table S3	NEW Table S3 for single-cell RNA-seq	R2 (comment 6)

DNA methylation at highlighted Bmp10 enhancer region

Figure A1. The quantification of DNA methylation level at the *Bmp10* enhancers. The averaged DNA methylation levels within the potential *Bmp10* enhancers were shown (highlighted in red in Figure 1B, bottom panel).

	E12.5		E15.5	
FPKM	Ctr	DKO	Ctr	DKO
Tbx20	191.0	201.0	177.0	205.0
Ttn	26.7	15.4	39.0	21.0

Figure A2. The RNA-seq FPKM values of *Tbx20* and *Ttn* in bulk heart tissue collected from control and Tet2/3-DKO embryos at the indicated developmental stages (E12.5 and E15.5). *Tbx20* displayed no or slightly increased FPKM value in Tet2/3 DKO embryonic heart tissues compared with control; while the expression of *Ttn* was reduced for the Tet2/3 DKO group at either E12.5 or E15.5.

Figure A3. Immunoblotting analyses showing the co-immunoprecipitation (co-IP) of Tet1 and YY1 in mouse embryonic stem cells (mESC). The anti-YY1 antibody was used to perform IP and an anti-Tet1 antibody was used to detect the interaction between YY1 and Tet1. IgG was used as control. The experiment was repeated 3 times.

Figure A4. Comparison of YY1 CUT&RUN with ATAC-seq and DNA methylation analysis in E12.5 mouse embryonic heart tissues. Differential YY1 enriched regions, chromatin accessibility regions and DNA methylated regions (DMR) were identified from YY1 CUT&RUN, ATAC-seq and WGBS data between the control and Tet2/3 DKO groups (E12.5). The overlapping differential YY1 enriched regions were compared with differential chromatin accessible regions (ATAC-seq data) (left), or DMR (right).

Figure A5. The complete list of GREAT analysis results:

Figure 1C HyperDMRs

Figure A5. The complete list of GREAT analysis results:

Figure 1C HypoDMRs

Figure A5. The complete list of GREAT analysis results:

Figure 4D (left panel)

Figure A5. The complete list of GREAT analysis results:

Figure 4D (right panel)

Figure A5. The complete list of GSEA analysis results

NEW Figure 7D (original Figure 6D) GSEA analysis

Description	P-value
Hypertrophic cardiomyopathy (HCM)	5.26E-11
Arrhythmogenic right ventricular cardiomyopathy (ARVC)	2.55E-10
Dilated cardiomyopathy	2.08E-09
Pathways in cancer	2.33E-07
Adherens junction	9.90E-06
Cardiac muscle contraction	1.52E-05
Focal adhesion	3.61E-05
ECM-receptor interaction	1.93E-04
Cytokine-cytokine receptor interaction	3.68E-04
Cell adhesion molecules (CAMs)	3.97E-04
Purine metabolism	1.09E-03
Axon guidance	1.86E-03
Calcium signaling pathway	2.09E-03
Basal cell carcinoma	2.17E-03
Hedgehog signaling pathway	2.32E-03
Melanogenesis	3.53E-03

REVIEWERS' COMMENTS:

Reviewer #1 (Remarks to the Author):

The authors have adequately addressed my comments.

Reviewer #2 (Remarks to the Author):

The authors provided a revised manuscript. They included a substantial amount of new data. The manuscript reads well. Although they did not perform all the experiments requested - their new data supports their interpretations well. I support proceeding with publication. Of minor note, the LB dataset the authors referenced is from murine ESCs and ESC-derived CMs, not E12.5 CMs.

Reviewer #3 (Remarks to the Author):

The authors have addressed all of my concerns.

Jonathan Brow

Specific points:

Reviewer #2 (Remarks to the Author):

The authors provided a revised manuscript. They included a substantial amount of new data. The manuscript reads well. Although they did not perform all the experiments requested – their new data supports their interpretations well. I support proceeding with publication. Of minor note, the LB dataset the authors referenced is from murine ESCs and ESC-derived CMs, not E12.5 CMs.

We have corrected the cell types referred in the following paragraph.

We thank the reviewer for pointing out the connection between YY1 and lamina associated chromatin regions. Since the current study is primarily focused on studying how TET/5hmC regulates YY1 binding, our logic is to first analyze the association between 5hmC and lamina-associated regions. Our assumption is that if we can identify 5hmC-enriched lamina-associated regions, we will move on to explore the regulation of YY1 within these regions. We analyzed the 5hmC peaks identified from our E12.5 CM within lamina-enriched regions using the published lamin B ChIP-seq data (GSE97878) in ESC differentiated cardiac progenitors. Unfortunately, there is no overlap between lamin B (n = 120,000) enriched regions and reduced 5hmC peaks (Tet2/3-DKO vs Control) (n = 8,846) identified from Tet2/3-DKO E12.5 heart tissue. This is largely consistent with our observation that 5hmC is enriched in compartment A, but not in compartment B. Based on this finding, it is less likely that YY1 is involved in instructing lamina-associated regions in our experiments. However, since only 16% of YY1 enriched regions overlapped with 5hmC peaks, we still cannot rule out the possibility that other YY1-bound regions (not overlapped with 5hmC enriched regions) might be involved in regulating A-to-B switch; but this topic is beyond the scope of this study.